# LEARNING VARIATIONAL NEIGHBOR LABELS FOR TEST-TIME DOMAIN GENERALIZATION

## ABSTRACT

This paper strives for domain generalization, where models are trained exclusively on source domains before being deployed on unseen target domains. We follow the strict separation of source training and target testing, but exploit the value of the unlabeled target data itself during inference. We make three contributions. First, we propose probabilistic pseudo-labeling of target samples to generalize the source-trained model to the target domain at test time. We formulate the generalization at test time as a variational inference problem, by modeling pseudo labels as distributions, to consider the uncertainty during generalization and alleviate the misleading signal of inaccurate pseudo labels. Second, we learn variational neighbor labels that incorporate the information of neighboring target samples to generate more robust pseudo labels. Third, to learn the ability to incorporate more representative target information and generate more precise and robust variational neighbor labels, we introduce a meta-generalization stage during training to simulate the generalization procedure. Experiments on seven widely-used datasets demonstrate the benefits, abilities, and effectiveness of our proposal.

## 1 INTRODUCTION

As soon as test data distributions differ from the ones experienced during training, deep neural networks start to exhibit generalizability problems and accompanying performance degradation (Geirhos et al., 2018; Recht et al., 2019). To deal with distribution shifts, domain generalization (Li et al., 2017; 2020; Motiian et al., 2017b; Muandet et al., 2013) has emerged as a promising tactic for generalizability to unseen target domains. However, as methods are only trained on source domains, this may still lead to overfitting and limited performance guarantees on unseen target domains.

To better adapt models to target domains – without relying on target data during training – test-time adaptation (Liang et al., 2023; Sun et al., 2020; Varsavsky et al., 2020; Wang et al., 2021) was introduced. It provides an alternative learning paradigm by training a model on source data and further adjusting the model according to the unlabeled target data at test time. Different settings for test-time adaptation have emerged. Test-time training (Sun et al., 2020) and test-time adaptation (Wang et al., 2021) attack image corruptions with a model trained on the original uncorrupted image distribution. The trained model is fine-tuned with self-supervised learning or entropy minimization to adapt to different corruptions in an online manner. The paradigm is also employed under the domain generalization setting using multiple source domains during training (Dubey et al., 2021; Iwasawa & Matsuo, 2021; Jang et al., 2023; Xiao et al., 2022), where the domain shifts are typically manifested in varying image styles and scenes, rather than corruptions. In this paper, we focus on the latter setting and refer to it as test-time domain generalization.

One widely applied strategy for updating models at test time is by optimizing or adjusting the model with target pseudo labels based on the source-trained model (Iwasawa & Matsuo, 2021; Jang et al., 2023). However, due to domain shifts, the source-model predictions of the target samples can be uncertain and inaccurate, leading to updated models that are overconfident on mispredictions (Yi et al., 2023). As a result, the obtained model becomes unreliable and misspecified to the target data (Wilson & Izmailov, 2020). In this paper, we make three contributions to attack the unreliability of test-time domain generalization by pseudo labels.

First, we define pseudo labels as stochastic variables and estimate their distributions. By doing so, the uncertainty in predictions of the source-trained model is incorporated into the generalization to the target data at test time, alleviating the misleading effects of uncertain and inaccurate pseudo labels. Second, due to the proposed probabilistic formalism, it is natural and convenient to utilize variational distributions to leverage extra information. By hinging on this benefit, we design variational neighbor labels that leverage the neighboring information of target samples into the inference of the pseudo-label distributions. This makes the variational labels more accurate, which enables the source-trained model to be better specified to target data and therefore conducive to model generalization on the target domain. Third, to learn the ability to incorporate more representative target information in the variational neighbor labels, we simulate the test-time generalization procedure across domains by meta-learning. Beyond the well-known meta-source and meta-target stages (Alet et al., 2021; Dou et al., 2019; Xiao et al., 2022), we introduce a meta-generalization stage in between the meta-source and meta-target stages to mimic the target generalization procedure. Based on the multiple source domains seen during training, the model is exposed to different domain shifts iteratively and optimized to learn the ability to generalize to unseen domains. Our experiments on seven widely-used domain generalization benchmarks demonstrate the promise and effectiveness of our proposal.

## 2 RELATED WORK

**Domain generalization.** Domain generalization is introduced to learn a model on one or several source domains that can generalize well on any out-of-distribution target domain (Blanchard et al., 2011; Muandet et al., 2013; Zhou et al., 2022). Different from domain adaptation (Long et al., 2015; Luo et al., 2020; Wang & Deng, 2018), domain generalization methods do not access any target data during training. One of the most widely-used methods for domain generalization is domain-invariant learning (Arjovsky et al., 2019; Ghifary et al., 2016; Li et al., 2018c; Motiian et al., 2017a; Muandet et al., 2013; Zhao et al., 2020), which learns invariant feature representations across source domains. As an alternative, source domain augmentation methods (Li et al., 2018a; Qiao et al., 2020; Shankar et al., 2018; Zhou et al., 2020a;b) try to generate more source domains during training. Recently, meta-learning-based methods (Balaji et al., 2018; Chen et al., 2023a; Dou et al., 2019; Du et al., 2020; Li et al., 2018b) have been explored to learn the ability to handle domain shifts.

**Test-time adaptation.** Another solution to address distribution shifts without target data during training is adapting the model at test time. Source-free adaptation (Eastwood et al., 2021; Liang et al., 2020; Litrico et al., 2023) adapts the source-trained model to the entire target set. Differently, test-time adaptation achieves adaptation and prediction in an online manner, without halting inference. One common test-time adaptation is fine-tuning by entropy minimization (Wang et al., 2021; Goyal et al., 2022; Jang et al., 2023; Niu et al., 2022; Zhang et al., 2022). Since entropy minimization does not consider the uncertainty of source model predictions, probabilistic algorithms (Brahma & Rai, 2022; Zhou & Levine, 2021) based on Bayesian semi-supervised learning and models fine-tuned on soft pseudo labels (Rusak et al., 2021; Zou et al., 2019) have been proposed. Different from these works, we introduce the uncertainty by considering pseudo labels as latent variables and estimate their distributions by variational inference. Our models consider uncertainty within the same probabilistic framework, without introducing extra models or knowledge distillation operations.

**Test-time domain generalization.** Many test-time adaptation methods adjust models to corrupted data distributions with a single source distribution during training (Sun et al., 2020; Wang et al., 2021). The idea of adjusting the source-trained model at test time is further explored under the domain generalization setting to consider target information for better generalization (Dubey et al., 2021; Iwasawa & Matsuo, 2021; Xiao et al., 2023; Zhang et al., 2021). We refer to these methods as test-time domain generalization. Dubey et al. (2021) generate domain-specific classifiers for the target domain with the target domain embeddings. Iwasawa & Matsuo (2021) adjust their prototypical classifier online according to the pseudo labels of the target data. Some also investigated meta-learning for test-time domain generalization (Alet et al., 2021; Du et al., 2021; Xiao et al., 2022). These methods mimic domain shifts during training with multiple source domains. Du et al. (2021) meta-learn to estimate the batch normalization statistics from each target sample to adjust the source-trained model. Xiao et al. (2022) learn to adapt their classifier to each individual target sample by mimicking domain shifts during training. Our method also learns the ability to adjust the model by unseen data under the multi-source meta-learning setting. Differently, we design meta-generalization and meta-target stages

during training to simulate both the generalization and inference procedures at test time. Our entire algorithm is explored under a probabilistic framework.

**Pseudo-label learning.** Pseudo-label learning relies on model predictions for retraining on downstream tasks. It is often applied for unlabeled data and self-training (Li et al., 2022; Miyato et al., 2018; Xie et al., 2020; Yalniz et al., 2019). To better utilize information from unlabeled target distributions, pseudo labels are also beneficial for unsupervised domain adaptation (Liu et al., 2021a; Shu et al., 2018; Zou et al., 2019), test-time adaptation (Chen et al., 2022; Rusak et al., 2021; Wang et al., 2022), and test-time domain generalization (Iwasawa & Matsuo, 2021; Jang et al., 2023; Wang et al., 2023). As pseudo labels can be noisy and overconfident (Zou et al., 2019), several studies focus on the appropriate selection and uncertainty of the pseudo labels. These works either select the pseudo labels with criteria such as the entropy consistency score of model predictions. (Liu et al., 2021a; Niu et al., 2022; Shin et al., 2022) or use soft pseudo labels to take the uncertainty into account (Rusak et al., 2021; Yang et al., 2022; Zou et al., 2019). We also use pseudo labels to generalize the source-trained model to the target domain. Different from the previous methods, we are the first to introduce pseudo labels as latent variables in a probabilistic parameterized framework for test-time domain generalization, where we incorporate uncertainty and generate pseudo labels with neighboring information through variational inference and meta-learning.

## 3 METHODOLOGY

**Preliminary.** We are given data from different domains defined on the joint space $\mathcal{X} \times \mathcal{Y}$, where $\mathcal{X}$ and $\mathcal{Y}$ denote the data space and label space, respectively. The domains are split into several source domains $\mathcal{D}_s = \left\{ (\mathbf{x}_s, \mathbf{y}_s)^i \right\}_{i=1}^{N_s}$ and the target domain $\mathcal{D}_t = \left\{ (\mathbf{x}_t, \mathbf{y}_t)^i \right\}_{i=1}^{N_t}$. Our goal is to train a model on source domains that is expected to generalize well on the (unseen) target domain.

We follow the test-time domain generalization setting (Dubey et al., 2021; Iwasawa & Matsuo, 2021; Xiao et al., 2022), where a source-trained model is generalized to target domains by adjusting the model parameters at test time. A common strategy for adjusting the model parameters is that the model $\boldsymbol{\theta}$ is first trained on source data $\mathcal{D}_s$ by minimizing a supervised cross-entropy ($L_{CE}$) loss $\mathcal{L}_{train}(\boldsymbol{\theta}) = \mathbb{E}_{(\mathbf{x}_s, \mathbf{y}_s)^i \in \mathcal{D}_s}[L_{CE}(\mathbf{x}_s, \mathbf{y}_s; \boldsymbol{\theta})]$; and then at test time the source-trained model $\boldsymbol{\theta}_s$ is generalized to the target domain by optimization with certain surrogate losses, e.g., entropy minimization ($L_E$), based on the online unlabeled test data, which is formulated as:

$$\mathcal{L}_{test}(\boldsymbol{\theta}) = \mathbb{E}_{\mathbf{x}_t \in \mathcal{D}_t}[L_E(\mathbf{x}_t; \boldsymbol{\theta}_s)], \tag{1}$$

where the entropy is calculated on the source model predictions. However, test samples from the target domain could be largely misclassified by the source model due to the domain shift, resulting in large uncertainty in the predictions. Moreover, the entropy minimization tends to update the model with high confidence even for the wrong predictions, which would cause a misspecified model for the target domain. To solve those problems, we address test-time domain generalization from a probabilistic perspective and further propose variational neighbor labels to incorporate more target information. A graphical illustration to highlight the differences between common test-time domain generalization and our proposals is shown in Figure 1.

**Probabilistic pseudo-labeling.** Given target sample $\mathbf{x}_t$ and source-trained model $\boldsymbol{\theta}_s$, we would like to make predictions on the target sample, formulated as $p(\mathbf{y}_t|\mathbf{x}_t, \boldsymbol{\theta}_s) = \int p(\mathbf{y}_t|\mathbf{x}_t, \boldsymbol{\theta}_t)p(\boldsymbol{\theta}_t|\mathbf{x}_t, \boldsymbol{\theta}_s)d\boldsymbol{\theta}_t$. Since the distribution of $p(\boldsymbol{\theta}_t)$ is intractable, the common test-time adaptation and generalization methods usually optimize the source model to the target one by the maximum a posterior (MAP), which is an empirical Bayesian method and an approximation of the integration of $p(\boldsymbol{\theta}_t)$ (Finn et al., 2018). The predictive likelihood is then formulated as:

$$p(\mathbf{y}_t|\mathbf{x}_t, \boldsymbol{\theta}_s) = \int p(\mathbf{y}_t|\mathbf{x}_t, \boldsymbol{\theta}_t)p(\boldsymbol{\theta}_t|\mathbf{x}_t, \boldsymbol{\theta}_s)d\boldsymbol{\theta}_t \approx p(\mathbf{y}_t|\mathbf{x}_t, \boldsymbol{\theta}_t^*), \tag{2}$$

where $\boldsymbol{\theta}_t^*$ is the MAP value of the optimized target model. The MAP approximation is interpreted as inferring the posterior over $\boldsymbol{\theta}_t$: $p(\boldsymbol{\theta}_t|\mathbf{x}_t, \boldsymbol{\theta}_s) \approx \delta(\boldsymbol{\theta}_t = \boldsymbol{\theta}_t^*)$, following a Dirac delta distribution.

To model the uncertainty of predictions for more robust test-time generalization, we treat pseudo labels as stochastic variables in the probabilistic framework of common test-time generalization as shown in Figure 1 (b). The pseudo labels are obtained from the source model predictions, which

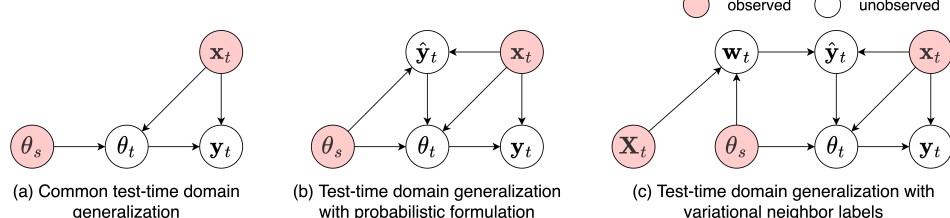

Figure 1: **Probabilistic modeling graph for test-time domain generalization.** (a) Common test-time domain generalization algorithm obtains the target model $\boldsymbol{\theta}_t$ by self-learning of the unlabeled target data $\mathbf{x}_t$ on source-trained model $\boldsymbol{\theta}_s$ (Iwasawa & Matsuo, 2021; Jang et al., 2023). (b) We introduce pseudo labels $p(\hat{\mathbf{y}}_t)$ as a latent variable to generate $p(\boldsymbol{\theta}_t)$ for more robust generalization. (c) We further propose variational neighbor labels to incorporate neighboring information into the generation of pseudo labels, where latent variable $\mathbf{w}_t$ and $\hat{\mathbf{y}}_t$ follow Gaussian and categorical distributions.

follow categorical distributions. Then we reformulate eq. (2) as:

$$
\begin{aligned}
p(\mathbf{y}_t|\mathbf{x}_t, \boldsymbol{\theta}_s) &= \int p(\mathbf{y}_t|\mathbf{x}_t, \boldsymbol{\theta}_t) \Big[ \int p(\boldsymbol{\theta}_t|\hat{\mathbf{y}}_t, \mathbf{x}_t, \boldsymbol{\theta}_s) p(\hat{\mathbf{y}}_t|\mathbf{x}_t, \boldsymbol{\theta}_s) d\hat{\mathbf{y}}_t \Big] d\boldsymbol{\theta}_t \\
&\approx \mathbb{E}_{p(\hat{\mathbf{y}}_t|\mathbf{x}_t, \boldsymbol{\theta}_s)}[p(\mathbf{y}_t|\mathbf{x}_t, \boldsymbol{\theta}_t^*)],
\end{aligned}
\tag{3}
$$

where $\boldsymbol{\theta}_t^*$ is the MAP value of $p(\boldsymbol{\theta}_t|\hat{\mathbf{y}}_t, \mathbf{x}_t, \boldsymbol{\theta}_s)$, obtained via gradient descent on the data $\mathbf{x}_t$ and the corresponding pseudo labels $\hat{\mathbf{y}}_t$ starting from $\boldsymbol{\theta}_s$. Note that we only use MAP approximation with gradient descent to estimate the model parameter $\boldsymbol{\theta}_t$, which will not hurt the generation of the probabilistic pseudo labels. This formulation allows us to sample different pseudo labels from the categorical distribution $p(\hat{\mathbf{y}}_t)$ to update the model $\boldsymbol{\theta}_t^*$, which takes into account the uncertainty of the source-trained predictions.

The common pseudo-labeling method can be treated as a specific case of eq. 3, which approximates the expectation of $p(\hat{\mathbf{y}}_t)$ by utilizing the `argmax` function on $p(\hat{\mathbf{y}}_t)$, generating the hard pseudo labels. $\boldsymbol{\theta}_t^*$ is then obtained by a point estimation of the hard pseudo labels. However, due to domain shifts, the `argmax` value of $p(\hat{\mathbf{y}}_t)$ is not guaranteed to always be correct. The optimization of the source-trained model then is similar to entropy minimization (eq. 1), where the updated model can achieve high confidence but wrong predictions of some target samples due to domain shifts. More analysis is provided in Appendix A.

**Variational neighbor labels.** To optimize the probabilistic framework, we use variational inference to approximate the true posterior of the probabilistic pseudo labels, in which we introduce more neighboring target information and categorical information during training. On one hand, introducing variational inference into pseudo-labeling is natural and convenient under the proposed probabilistic formulation. On the other hand, to generate pseudo labels that are more accurate and calibrated for more robust generalization, it is necessary to incorporate more target information. Assume that we have a mini-batch of target data $\mathbf{X}_t = \{\mathbf{x}_t^i\}_{i=1}^M$, we reformulate eq. (3) as:

$$
\begin{aligned}
p(\mathbf{y}_t|\mathbf{x}_t, \boldsymbol{\theta}_s, \mathbf{X}_t) &= \int p(\mathbf{y}_t|\mathbf{x}_t, \boldsymbol{\theta}_t) \Big[ \int \int p(\boldsymbol{\theta}_t|\hat{\mathbf{y}}_t, \mathbf{x}_t, \boldsymbol{\theta}_s) p(\hat{\mathbf{y}}_t, \mathbf{w}_t|\mathbf{x}_t, \boldsymbol{\theta}_s, \mathbf{X}_t) d\hat{\mathbf{y}}_t d\mathbf{w}_t \Big] d\boldsymbol{\theta}_t \\
&= \int \int p(\mathbf{y}_t|\mathbf{x}_t, \boldsymbol{\theta}_t^*) p(\hat{\mathbf{y}}_t, \mathbf{w}_t|\mathbf{x}_t, \boldsymbol{\theta}_s, \mathbf{X}_t) d\hat{\mathbf{y}}_t d\mathbf{w}_t.
\end{aligned}
\tag{4}
$$

As in eq. (3), $\boldsymbol{\theta}_t^*$ is the MAP value of $p(\boldsymbol{\theta}_t|\hat{\mathbf{y}}_t, \mathbf{x}_t, \boldsymbol{\theta}_s)$. We introduce the latent variable $\mathbf{w}_t$ to integrate the information of the neighboring target samples $\mathbf{X}_t$ as shown in Figure 1 (c). To facilitate the estimation of the variational neighbor labels, we set the prior distribution as:

$$
p(\hat{\mathbf{y}}_t, \mathbf{w}_t|\mathbf{x}_t, \boldsymbol{\theta}_s, \mathbf{X}_t) = p(\hat{\mathbf{y}}_t|\mathbf{w}_t, \mathbf{x}_t) p_\phi(\mathbf{w}_t|\boldsymbol{\theta}_s, \mathbf{X}_t),
\tag{5}
$$

where $p_\phi(\mathbf{w}_t|\boldsymbol{\theta}_s, \mathbf{X}_t)$ is generated by the features of $\mathbf{X}_t$ together with their output values based on $\boldsymbol{\theta}_s$. In detail, to explore the information of neighboring target samples, we first generate the predictions of $\mathbf{X}_t$ by the source-trained model $\boldsymbol{\theta}_s$. Then we estimate the averaged target features of each category according to the source-model predictions. The latent variable $\mathbf{w}_t$ is obtained by the model $\phi$ with the averaged features as the input. Therefore, $\mathbf{w}_t$ contains the categorical information of the target features and can be treated as an updated classifier with more target information. The variational neighbor labels $\hat{\mathbf{y}}_t$ are obtained by classifying the target samples using $\mathbf{w}_t$. Rather than directly using

the source model $\boldsymbol{\theta}_s$, we estimate $\hat{\mathbf{y}}_t$ from the latent variable $\mathbf{w}_t$, which integrates the information of neighboring target samples to be more accurate and reliable.

To approximate the true posterior of the joint distribution $p(\hat{\mathbf{y}}_t, \mathbf{w}_t)$ and incorporate more representative target information, we design a variational posterior $q(\hat{\mathbf{y}}_t, \mathbf{w}_t | \mathbf{x}_t, \boldsymbol{\theta}_s, \mathbf{X}_t, \mathbf{Y}_t)$ to supervise the prior distribution $p(\hat{\mathbf{y}}_t, \mathbf{w}_t | \mathbf{x}_t, \boldsymbol{\theta}_s, \mathbf{X}_t)$ during training:

$$q(\hat{\mathbf{y}}_t, \mathbf{w}_t | \mathbf{x}_t, \boldsymbol{\theta}_s, \mathbf{X}_t, \mathbf{Y}_t) = p(\hat{\mathbf{y}}_t | \mathbf{w}_t, \mathbf{x}_t) q_{\boldsymbol{\phi}}(\mathbf{w}_t | \boldsymbol{\theta}_s, \mathbf{X}_t, \mathbf{Y}_t). \tag{6}$$

The variational posterior distribution is obtained similarly as the prior by generating $\mathbf{w}_t$ through the categorical averaged features. The model $\boldsymbol{\phi}$ is shared by the prior and posterior distributions. The main difference is that the averaged features to generate $\mathbf{w}_t$ are obtained with the actual target labels $\mathbf{Y}_t$. Since the target labels $\mathbf{Y}_t$ are inaccessible, we can only utilize the prior distribution $p(\hat{\mathbf{y}}_t, \mathbf{w}_t | \mathbf{x}_t, \boldsymbol{\theta}_s, \mathbf{X}_t)$ at test time. Therefore, we introduce the variational posterior under the meta-learning framework (Du et al., 2021; Finn et al., 2017; Xiao et al., 2022), where we mimic domain shifts and the test-time generalization procedure during training to learn the variational neighbor labels. In this case, according to the variational posterior distribution, the prior distribution $p(\hat{\mathbf{y}}_t | \mathbf{w}_t, \mathbf{x}_t) p_{\boldsymbol{\phi}}(\mathbf{w}_t | \boldsymbol{\theta}_s, \mathbf{X}_t)$ learns the ability to incorporate more representative target information and generate more accurate neighbor labels.

**Meta-generalization with variational neighbor labels.** We split the source domains $\mathcal{D}_s$ into meta-source domains $\mathcal{D}_{s'}$ and a meta-target domain $\mathcal{D}_{t'}$ during training. The meta-target domain is selected randomly in each iteration to mimic diverse domain shifts. Moreover, we divide each iteration into meta-source, meta-generalization, and meta-target stages to simulate the training stage on source domains, test-time generalization, and test stage on target data, respectively.

*Meta-source.* We train the meta-source model $\boldsymbol{\theta}_{s'}$ by minimizing the supervised loss $L_{\text{CE}}(\mathbf{x}_{s'}, \mathbf{y}_{s'}; \boldsymbol{\theta})$, where $(\mathbf{x}_{s'}, \mathbf{y}_{s'})$ denotes the input-label sample pairs of the meta-source domains.

*Meta-generalization.* To mimic test-time generalization and prediction, our goal in the newly introduced meta-generalization stage is to optimize the meta-source model $\boldsymbol{\theta}_{s'}$ by the meta-target data and make predictions with the generalized model. By introducing the variational neighbor labels, the log-likelihood of the meta-target prediction $\mathbf{y}_{t'}$ is formulated as:

$$p(\mathbf{y}_{t'} | \mathbf{x}_{t'}, \boldsymbol{\theta}_{s'}, \mathbf{X}_{t'}) = \int \int p(\mathbf{y}_{t'} | \mathbf{x}_{t'}, \boldsymbol{\theta}_{t'}^*) p(\hat{\mathbf{y}}_{t'}, \mathbf{w}_{t'} | \mathbf{x}_{t'}, \boldsymbol{\theta}_{s'}, \mathbf{X}_{t'}) d\hat{\mathbf{y}}_{t'} d\mathbf{w}_{t'}, \tag{7}$$

where $\boldsymbol{\theta}_{t'}^*$ is the MAP value of $p(\boldsymbol{\theta}_{t'} | \hat{\mathbf{y}}_{t'}, \mathbf{x}_{t'}, \boldsymbol{\theta}_{s'})$, similar to eq. (4), and $p(\hat{\mathbf{y}}_{t'}, \mathbf{w}_{t'} | \mathbf{x}_{t'}, \boldsymbol{\theta}_{s'}, \mathbf{X}_{t'}) = p(\hat{\mathbf{y}}_{t'} | \mathbf{w}_{t'}, \mathbf{x}_{t'}) p_{\boldsymbol{\phi}}(\mathbf{w}_{t'} | \boldsymbol{\theta}_{s'}, \mathbf{X}_{t'})$ is the joint prior distribution of the meta-target neighbor labels $\hat{\mathbf{y}}_{t'}$ and latent variable $\mathbf{w}_{t'}$. The joint variational posterior is designed as $q(\hat{\mathbf{y}}_{t'}, \mathbf{w}_{t'} | \mathbf{x}_{t'}, \boldsymbol{\theta}_{s'}, \mathbf{X}_{t'}, \mathbf{Y}_{t'}) = p(\hat{\mathbf{y}}_{t'} | \mathbf{w}_{t'}, \mathbf{x}_{t'}) q_{\boldsymbol{\phi}}(\mathbf{w}_{t'} | \boldsymbol{\theta}_{s'}, \mathbf{X}_{t'}, \mathbf{Y}_{t'})$ to learn more reliable neighbor labels by considering the actual labels $\mathbf{Y}_{t'}$ of the meta-target data. Under this meta-learning setting, the actual labels $\mathbf{Y}_{t'}$ of the meta-target data are accessible during source training. Thus, the variational distribution utilizes both the domain and categorical information of the neighboring samples and models the meta-target distribution more reliably, generating more accurate neighbor labels $\hat{\mathbf{y}}_{t'}$ of the meta-target samples. With the variational neighbor labels $\hat{\mathbf{y}}_{t'}$ the test-time domain generalization procedure is simulated by obtaining $\boldsymbol{\theta}_{t'}^*$ from:

$$\boldsymbol{\theta}_{t'}^* = \boldsymbol{\theta}_{s'} - \lambda_1 \nabla_{\boldsymbol{\theta}} L_{\text{CE}}(\mathbf{x}_{t'}, \hat{\mathbf{y}}_{t'}; \boldsymbol{\theta}_{s'}), \quad \hat{\mathbf{y}}_{t'} \sim p(\hat{\mathbf{y}}_{t'} | \mathbf{w}_{t'}, \mathbf{x}_{t'}), \quad \mathbf{w}_{t'} \sim q_{\boldsymbol{\phi}}(\mathbf{w}_{t'} | \boldsymbol{\theta}_{s'}, \mathbf{X}_{t'}, \mathbf{Y}_{t'}), \tag{8}$$

where $\lambda_1$ denotes the learning rate of the optimization in the meta-generalization stage.

*Meta-target.* Since our final goal is to obtain good performance on the target data after optimization, we further mimic the test-time inference on the meta-target domain and supervise the meta-target prediction on $\boldsymbol{\theta}_{t'}^*$ by maximizing the log-likelihood of eq (7):

$$\log p(\mathbf{y}_{t'} | \mathbf{x}_{t'}, \boldsymbol{\theta}_{s'}, \mathbf{X}_{t'}) \geq \mathbb{E}_{q_{\boldsymbol{\phi}}(\mathbf{w}_{t'})} \mathbb{E}_{p(\hat{\mathbf{y}}_{t'} | \mathbf{w}_{t'}, \mathbf{x}_{t'})} [\log p(\mathbf{y}_{t'} | \mathbf{x}_{t'}, \boldsymbol{\theta}_{t'}^*)] - \mathbb{D}_{KL}[q_{\boldsymbol{\phi}}(\mathbf{w}_{t'}) || p_{\boldsymbol{\phi}}(\mathbf{w}_{t'})], \tag{9}$$

where $p_{\boldsymbol{\phi}}(\mathbf{w}_{t'}) = p_{\boldsymbol{\phi}}(\mathbf{w}_{t'} | \boldsymbol{\theta}_{s'}, \mathbf{X}_{t'})$ generated by the features of $\mathbf{X}_{t'}$ together with their output values based on $\boldsymbol{\theta}_{s'}$. $q_{\boldsymbol{\phi}}(\mathbf{w}_{t'}) = q_{\boldsymbol{\phi}}(\mathbf{w}_{t'} | \boldsymbol{\theta}_{s'}, \mathbf{X}_{t'}, \mathbf{Y}_{t'})$ is obtained by the features of $\mathbf{X}_{t'}$ considering the actual labels $\mathbf{Y}_{t'}$. The detailed formulation is provided in Appendix A.

As aforementioned, the actual labels $\mathbf{y}_{t'}$ of the meta-target data are accessible during training. We can further supervise the updated model $\boldsymbol{\theta}_{t'}^*$ on its meta-target predictions by the actual labels. Maximizing the log-likelihood $\log p(\mathbf{y}_{t'} | \mathbf{x}_{t'}, \boldsymbol{\theta}_{s'}, \mathbf{X}_{t'})$ is equal to minimizing:

$$\mathcal{L}_{meta} = \mathbb{E}_{(\mathbf{x}_{t'}, \mathbf{y}_{t'})} [\mathbb{E}_{q_{\boldsymbol{\phi}}(\mathbf{w}_{t'})} \mathbb{E}_{p(\hat{\mathbf{y}}_{t'} | \mathbf{w}_{t'}, \mathbf{x}_{t'})} L_{\text{CE}}(\mathbf{x}_{t'}, \mathbf{y}_{t'}; \boldsymbol{\theta}_{t'}^*)] + \mathbb{D}_{KL}[q_{\boldsymbol{\phi}}(\mathbf{w}_{t'}) || p_{\boldsymbol{\phi}}(\mathbf{w}_{t'})]. \tag{10}$$

The source model $\boldsymbol{\theta}_s$ in each iteration is finally updated by $\boldsymbol{\theta}_s = \boldsymbol{\theta}_{s'} - \lambda_2 \nabla_{\boldsymbol{\theta}} \mathcal{L}_{meta}$, where $\lambda_2$ denotes the learning rate for the meta-target stage. Note that the loss in eq. (10) is computed on the $\boldsymbol{\theta}_{t'}^*$ obtained by eq. (8), while the optimization is performed over the meta-source model $\boldsymbol{\theta}_{s'}$. Intuitively, the model updated by the meta-target neighbor labels is trained to achieve good performance on the meta-target data. Thus, the meta-generalization stage is further supervised to optimize the model well across domains and better generate and utilize the variational neighbor labels.

The variational inference model $\phi$ is also optimized in the meta-target stage. To guarantee that the variational neighbor labels do extract the categorical neighboring information for classification, we add an extra cross-entropy loss ($L_{CE}$) on the variational neighbor labels with actual labels during the meta-target stage. Thus, $\phi$ is updated with a learning rate $\lambda_3$ by $\phi = \phi - \lambda_3 (\nabla_\phi L_{CE} + \nabla_\phi \mathcal{L}_{meta})$. By simulating distribution shifts during training, the model learns the ability to generate more effective pseudo labels for fine-tuning the model across distribution shifts. The variational neighbor labels are further improved by considering more neighboring target information.

**Test-time generalization.** At test time, the model trained on the source domains with the meta-learning strategy $\boldsymbol{\theta}_s$ is generalized to $\boldsymbol{\theta}_t^*$ by further optimization:

$$\boldsymbol{\theta}_t^* = \boldsymbol{\theta}_s - \lambda_1 \nabla_{\boldsymbol{\theta}} L_{\text{CE}}(\mathbf{x}_t, \hat{\mathbf{y}}_t; \boldsymbol{\theta}_s), \quad \hat{\mathbf{y}}_t \sim p(\hat{\mathbf{y}}_t | \mathbf{w}_t, \mathbf{x}_t), \quad \mathbf{w}_t \sim p_\phi(\mathbf{w}_t | \boldsymbol{\theta}_s, \mathbf{X}_t). \quad (11)$$

Since the target labels $\mathbf{Y}_t$ are inaccessible, we generate neighbor labels $\hat{\mathbf{y}}_t$ and latent variables $\mathbf{w}_t$ from the prior distribution $p(\hat{\mathbf{y}}_t, \mathbf{w}_t | \mathbf{x}_t, \boldsymbol{\theta}_s, \mathbf{X}_t) = p(\hat{\mathbf{y}}_t | \mathbf{w}_t, \mathbf{x}_t) p_\phi(\mathbf{w}_t | \boldsymbol{\theta}_s, \mathbf{X}_t)$. The distribution $p(\mathbf{w}_t)$ is inferred as a Gaussian distribution by generating the mean $\mu$ and variance $\sigma$ using the target averaged features through $\phi$. Then we sample $\mathbf{w}_t$ by Monte Carlo sampling and generate the categorical distribution $p(\hat{\mathbf{y}}_t)$ with the input target features, which we utilize to obtain the MAP value $\boldsymbol{\theta}_t^*$. From $\boldsymbol{\theta}_t^*$ we make predictions on the (unseen) target data $\mathcal{D}_t$, formulated as:

$$\begin{aligned}
p(\mathbf{y}_t | \mathbf{x}_t, \boldsymbol{\theta}_s, \mathbf{X}_t) &= \int p(\mathbf{y}_t | \mathbf{x}_t, \boldsymbol{\theta}_t) \Big[ \int p(\boldsymbol{\theta}_t | \hat{\mathbf{y}}_t, \mathbf{x}_t, \boldsymbol{\theta}_s) p(\hat{\mathbf{y}}_t, \mathbf{w}_t | \mathbf{x}_t, \boldsymbol{\theta}_s, \mathbf{X}_t) d\hat{\mathbf{y}}_t d\mathbf{w}_t \Big] d\boldsymbol{\theta}_t \\
&= \mathbb{E}_{p_\phi(\mathbf{w}_t)} \mathbb{E}_{p(\hat{\mathbf{y}}_t | \mathbf{w}_t, \mathbf{x}_t)} [\log p(\mathbf{y}_t | \mathbf{x}_t, \boldsymbol{\theta}_t^*)].
\end{aligned} \quad (12)$$

We provide both the training algorithm and test-time generalization algorithm in Appendix B.

## 4 EXPERIMENTS

**Seven datasets.** We demonstrate the effectiveness of our method on image classification problems and evaluate it on seven widely used domain generalization datasets. Namely, *PACS* (Li et al., 2017): 7 classes, 4 domains and 9,991 images. *VLCS* (Fang et al., 2013): 5 classes, 4 domains and 10,729 images. *Office-Home* (Venkateswara et al., 2017): 65 classes, 4 domains and 15,500 images. *TerraIncognita* (Beery et al., 2018): 10 classes, 4 domains and 24,778 images. *Mini DomainNet* (Zhou et al., 2021): 126 classes, 4 domains and 140,000 images. We follow training and validation split in (Li et al., 2017) and evaluate model according to "leave-one-out" protocol (Li et al., 2019; Carlucci et al., 2019). We also evaluate our method on the *Rotated MNIST* and *Fashion-MNIST* datasets following Piratla et al. (2020).

**Implementation details.** We utilize ResNet-18 for all our experiments and ablation studies and report the accuracies on ResNet-50 for comparison as well. We evaluate the method on the online test-time domain generalization setting (Iwasawa & Matsuo, 2021), we increment the target data iteratively and keep updating and evaluating the model. When we report an ERM baseline, it means we directly evaluate the source-trained model without any adjustment at test time (Gulrajani & Lopez-Paz, 2020). The backbones are pretrained on ImageNet same as the previous methods.

During training, we use a varied learning rate throughout the model and train the model for 10,000 iterations. In the meta-generalization procedure, we set the learning rate $\lambda_1$ as $1e-4$ for all layers. During meta-target, we set the learning rate for the pretrained ResNet ($\lambda_2$) to $5e-5$ and the learning rate of the variational module $\phi$ ($\lambda_3$) and classifiers as $1e-4$ for all datasets. The batch size is set to 70 during the training and set to 20 during the test-time generalization procedure. At test time, we use the learning rate of $1e-4$ for all the layers and update all parameters. All hyperparameters for source training and test-time using the training validation set have been selected as mentioned in (Iwasawa & Matsuo, 2021). We use similar settings and hyperparameters for all domain generalization benchmarks. The method introduces a small computational cost for inference

Table 1: **Ablations on variational neighbor labels**. Results on PACS and TerraIncognita with ResNet-18. Our probabilistic formulation performs better than the common pseudo-labeling baseline for test-time domain generalization by considering the uncertainty. Incorporating more target information by the variational neighbor labels improves results further, especially when used in concert with meta-generalization. We provide per-domain results in Appendix F.

|  |  | PACS | TerraIncognita |
|---|---|---|---|
| Pseudo-labeling baseline | (eq. 1) | $81.3 \pm 0.3$ | $41.2 \pm 0.4$ |
| Probabilistic pseudo-labeling | (eq. 3) | $82.0 \pm 0.2$ | $42.5 \pm 0.5$ |
| Variational neighbor-labeling | (eq. 4) | $82.4 \pm 0.3$ | $43.8 \pm 0.5$ |
| Meta-generalization with variational neighbor labels | (eq. 10) | $83.5 \pm 0.4$ | $46.2 \pm 0.6$ |

at test time and about 1% more parameters than the backbone model. The time cost for test-time generalization is competitive with other fine-tuning methods, with 5m 33s on PACS. We provide more implementation details and detailed computational costs in Appendix C.

**Ablations on variational neighbor labels.** To show the benefits of the proposed method, we conduct an ablation on PACS and TerraIncognita. We first compare the probabilistic pseudo-labeling (eq. 3) with the common one (eq. 1). As shown in the first two rows in Table 1, the probabilistic formulation performs better, which demonstrates the benefits of modeling uncertainty of the pseudo labels during generalization at test time. By incorporating more target information from the neighboring samples (eq. 4), the variational neighbor labels become more reliable, which benefits generalization on the target data. With the meta-generalization strategy (eq. 10), we learn the ability to incorporate more representative target information leading to further performance improvements. To show the benefits of meta-generalization, we conduct additional experiments for meta-generalization with pseudo-labeling and meta-generalization with probabilistic pseudo-labeling, which achieve 82.0 and 82.7 on PACS, respectively. Meta-generalization can also improve the other pseudo-labeling methods, while the proposed variational pseudo-labeling improves the most.

**Calibration ability.** To further show the benefits of the variational neighbor labels, we also investigate the calibration ability by measuring the Expected Calibration Error (Guo et al., 2017). We report hard and soft pseudo-labeling as baselines, as well as the results of the state-of-the-art method (Xiao et al., 2022) that considers uncertainty by variational inference. As shown in Figure 2, the error of our method is lower than the alternatives on all domains, demonstrating a better ability to model uncertainty at test time. By incorporating pseudo labels as latent variables with variational inference and considering neighboring target information, the proposed method models the uncertainty of the target samples more accurately. With the better-calibrated labels, the model achieves more robust generalization on the target domain at test time.

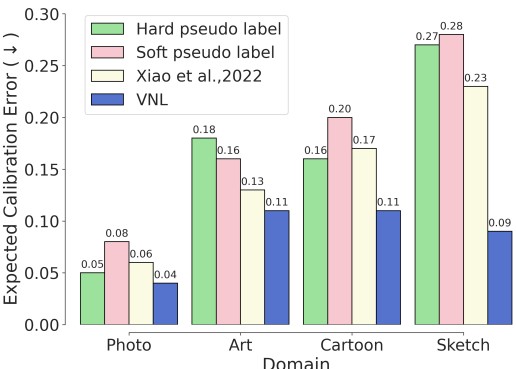

Figure 2: **Calibration ability** on PACS. Variational neighbor labels consistently have a lower Expected Calibration Error.

**Generalization in complex scenarios.** By considering the uncertainty and including more target information in the pseudo labels, our method can handle more complex test-time generalization scenarios. To demonstrate this ability, we conduct experiments with multiple target distributions on Rotated MNIST, as defined by Xiao et al. (2022). Specifically, we use 0°, 15°, 75° and 90° as source domains and 30°, 45° and 60° as target. As shown in Table 2, the common MAP method (Wang et al., 2021) achieves good results on the single target domains, while it is unable to outperform an ERM baseline on the multiple target domains. The proposed method per-

Table 2: **Generalization in complex scenarios.** Our method generalizes well on both single and multiple target distributions.

|  | Target distribution | |
|---|---|---|
|  | Single | Multiple |
| ERM baseline | 95.6 | 95.6 |
| Wang et al. (2021) | 96.5 | 95.6 |
| Xiao et al. (2022) | 96.9 | 96.9 |
| *VNL* | **97.5** $\pm 0.3$ | **97.4** $\pm 0.3$ |

forms well under both settings and better than Xiao et al. (2022), which achieves generalization on each sample, demonstrating the generalization ability of our method in more complex scenarios.

**Generalization with varying batch sizes.** Test-time generalization and adaptation methods usually require large batches of target samples to update the source-trained model. However, during real-world deployment, the number of available target samples may be limited. Thus constraining test-time generalization performance. In Figure 3 we compare with Tent (Wang et al., 2021) on PACS for varying batch sizes. Tent performs well with large batch sizes, but suffers with smaller batch sizes, e.g., 16, and is worse than the ERM baseline. By contrast, our method consistently achieves good results even with small target batch sizes. Demonstrating the benefit of incorporating the uncertainty and representative neighboring information. We provide more detailed results in Appendix F.

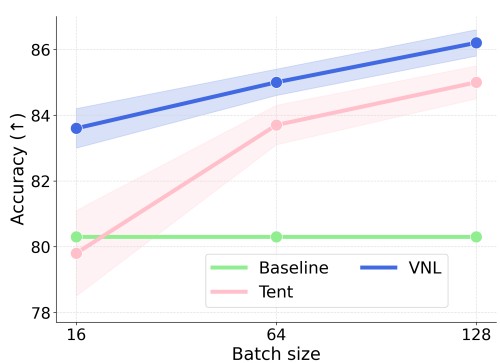

Figure 3: **Generalization with varying batch sizes.** Variational neighbor labels outperform Tent (Wang et al., 2021) on PACS, independent of batch size. Largest improvement for small batch sizes.

**Generalization along with inference.** For more insights into the variational neighbor labels, we provide the online performance along with generalization steps for the 'art' domain from PACS. As shown in Figure 4, starting from the same baseline accuracy, the gap between the results of variational neighbor labels and the hard pseudo labels becomes larger and larger along with the generalization steps. Variational neighbor labels achieve faster generalization of the source-trained model. After 50 iterations, the performance of the hard pseudo labels is saturated and even drops due to the error accumulation resulting from inaccurate pseudo labels during model updating. By considering the uncertainty and neighboring information, our variational neighbor labels improve performance and are less prone to saturation, leading to better accuracy.

**Orthogonality.** Since the proposed meta-learned variational neighbor labels focus on generating pseudo labels at test time, the method is orthogonal to other deployment techniques, e.g., data augmentation for generalization at test time (Zhang et al., 2022). Achieving test-time domain generalization compounded with these methods will further improve the performance. To demonstrate this, we conduct test-time generalization by our method with augmented target samples on PACS without altering the source training strategy. When adding similar augmentation as in (Zhang et al., 2022), we increase our results on ResNet-18 from 83.5% to 85.0% overall accuracy. We provide the complete table including the per-domain results in Appendix F. In the following, we report the results of our method in conjunction with augmentations.

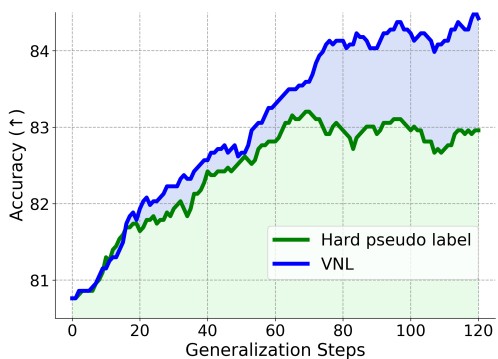

Figure 4: **Generalization along with inference.** Variational neighbor labels achieve faster generalization that is less prone to saturation.

**State-of-the-art comparisons.** We compare our proposal with state-of-the-art test-time domain generalization, as well as some standard domain generalization and test-time adaptation methods. Note the latter methods are designed for single-source image corruption settings, so we report the reimplemented results from Jang et al. (2023). Table 3 shows the results on PACS, VLCS, Office-Home, and TerraIncognita for both ResNet-18 and ResNet-50 backbones. Our method is competitive on most of the datasets, except for Office-Home where the sample-wise generalization of Xiao et al. (2022) performs better. The reason can be that the representative neighboring information is more difficult to incorporate with a larger number of categories (e.g., 65 in Office-Home), which needs larger capacity models $\phi$. We have experimented with $\phi$ values and obtained a mean accuracy of 57.1 with 2 layers and a mean accuracy of 64.3 with 3 layers in $\phi$.

Table 3: **State-of-the-art comparisons** for ResNet-18 (RN18) and ResNet-50 (RN50) backbones. Our results are averaged over five runs. Test-time adaptation results by Wang et al. (2021) and Liang et al. (2020) for domain generalization provided by Jang et al. (2023). Gray numbers for Xiao et al. (2022) based on our reimplementation. Our method is either best (bold) or runner-up (underlined).

| | PACS | | VLCS | | Office-Home | | TerraIncognita | |
|---|---|---|---|---|---|---|---|---|
| | RN18 | RN50 | RN18 | RN50 | RN18 | RN50 | RN18 | RN50 |
| **Standard domain generalization** | | | | | | | | |
| ERM baseline | 79.6 | 85.7 | 75.8 | 77.4 | 61.0 | 67.5 | 35.8 | 47.2 |
| Arjovsky et al. (2019) | 80.9 | 83.5 | 75.1 | 78.5 | 58.0 | 64.3 | 38.4 | 47.6 |
| Shi et al. (2022) | 82.0 | 85.5 | 76.9 | 77.8 | 62.0 | 68.6 | 40.2 | 45.1 |
| **Test-time adaptation on domain generalization** | | | | | | | | |
| Wang et al. (2021) | 83.9 | 85.2 | 72.9 | 73.0 | 60.9 | 66.3 | 33.7 | 37.1 |
| Liang et al. (2020) | 82.4 | 84.1 | 65.2 | 67.0 | 62.6 | 67.7 | 33.6 | 35.2 |
| **Test-time domain generalization** | | | | | | | | |
| Iwasawa & Matsuo (2021) | 81.7 | 85.3 | 76.5 | **80.0** | 57.0 | 68.3 | 41.6 | 47.0 |
| Dubey et al. (2021) | - | 84.1 | - | 78.0 | - | 67.9 | - | 47.3 |
| Jang et al. (2023) | 81.9 | 84.1 | 77.3 | 77.6 | 63.7 | 68.6 | 42.6 | 47.4 |
| Chen et al. (2023b) | 83.8 | - | 76.9 | - | 62.0 | - | 43.2 | - |
| Xiao et al. (2022) | 84.1 | 87.5 | 77.8 | 78.6 | **66.0** | **71.0** | 44.8 | 48.4 |
| *VNL* | **85.0** ±0.4 | **87.9** ±0.3 | **78.2** ±0.3 | 79.1 ±0.4 | 64.3 ±0.3 | 69.1 ±0.4 | **46.9** ±0.4 | **49.4** ±0.6 |

Note that our method still outperforms other recent methods (Chen et al., 2023b; Iwasawa & Matsuo, 2021; Jang et al., 2023; Wang et al., 2021) on Office-Home. Moreover, since we consider the uncertainty of the variational neighbor labels, the proposed method solves some hard cases of the single-sample approach reported in Xiao et al. (2022). As shown in Figure 5, our method has low confidence in the uncertain samples, e.g., with different objectives or limited information, showing good calibration of our method, which is also demon-

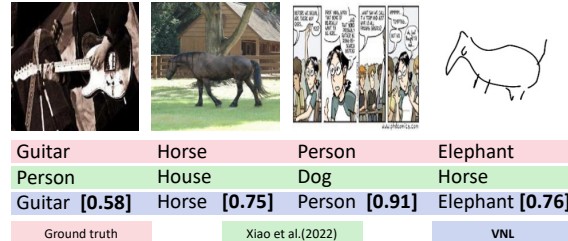

| Guitar | Horse | Person | Elephant |
|---|---|---|---|
| Person | House | Dog | Horse |
| Guitar **[0.58]** | Horse **[0.75]** | Person **[0.91]** | Elephant **[0.76]** |
| Ground truth | | Xiao et al.(2022) | VNL |

Figure 5: **Comparison on hard examples from Xiao et al. (2022)** on PACS. Our proposal is more robust on samples with multiple objectives or complex scenes.

strated in Figure 2. With the proposed method, the model predicts these hard cases correctly, showing the effectiveness of test-time generalization with the meta-generalized variational neighbor labels in complex scenes. In addition, there are also some recent standard domain generalization methods achieving good performance. For instance, (Gao et al., 2022) achieved good results on PACS, VLCS, Office-Home, and TerraIncognita based on ResNet-50 by utilizing an extra dataset before training to meta-learn loss function. This implies that we can also improve by utilizing more datasets during training. We provide more comparisons to the standard domain generalization methods in Appendix E. Experiments on Rotated MNIST, Fashion-MNIST, and mini-DomainNet are also provided in Appendix E. Our method also achieves competitive performance on these datasets.

**Limitations.** Since our method utilizes meta-learning and neighboring target information, it requires multiple source domains during training and small batches of target samples at test time, which can be a limitation in some environments. We consider a single-source and single-target-sample variant of our approach as a valuable investigation for future work.

## 5 CONCLUSION

We cast test-time domain generalization as a probabilistic inference problem and model pseudo labels as latent variables in the formulation. By incorporating the uncertainty of the pseudo labels, the probabilistic formulation mitigates updating the source-trained model with inaccurate supervision, which arises due to domain shifts and leads to misspecified models. Based on the probabilistic formulation, we further propose variational neighbor labels under the designed meta-generalization setting, which estimates the pseudo labels by incorporating neighboring target information through variational inference and learns the ability to generalize the source-trained model. Ablation studies and further comparisons show the benefits, abilities, and effectiveness of our method on seven common domain generalization datasets.

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

# A   DERIVATIONS

## A.1   DETAILED DERIVATION OF META-GENERALIZED VARIATIONAL NEIGHBOR LABELS

We start the objective function from $\log p(\mathbf{y}_{t'}|\mathbf{x}_{t'}, \boldsymbol{\theta}_{s'}, \mathbf{X}_{t'})$. Here we provide the detailed generating process of the formulation:

$$\log p(\mathbf{y}_{t'}|\mathbf{x}_{t'}, \boldsymbol{\theta}_{s'}, \mathbf{X}_{t'}) = \log \int p(\mathbf{y}_{t'}|\mathbf{x}_{t'}, \boldsymbol{\theta}_{t'})p(\boldsymbol{\theta}_{t'}|\mathbf{x}_{t'}, \boldsymbol{\theta}_{s'}, \mathbf{X}_{t'})d\boldsymbol{\theta}_{t'}. \tag{13}$$

We then introduce the pseudo labels $\hat{\mathbf{y}}_{t'}$ as the latent variable into eq. (13) and derive it as:

$$\log p(\mathbf{y}_{t'}|\mathbf{x}_{t'}, \boldsymbol{\theta}_{s'}, \mathbf{X}_{t'}) = \log \int p(\mathbf{y}_{t'}|\mathbf{x}_{t'}, \boldsymbol{\theta}_{t'}) \int p(\boldsymbol{\theta}_{t'}|\hat{\mathbf{y}}_{t'}, \mathbf{x}_{t'}, \boldsymbol{\theta}_{s'})p(\hat{\mathbf{y}}_{t'}|\mathbf{x}_{t'}, \boldsymbol{\theta}_{s'}, \mathbf{X}_{t'})d\hat{\mathbf{y}}_{t'}d\boldsymbol{\theta}_{t'}. \tag{14}$$

Theoretically, the distribution $p(\boldsymbol{\theta}_{t'})$ is obtained by $p(\boldsymbol{\theta}_{t'}|\mathbf{y}_{t'}, \mathbf{x}_{t'}, \boldsymbol{\theta}_{s'}) \propto p(\mathbf{y}_t|\mathbf{x}_t, \boldsymbol{\theta}_t)p(\boldsymbol{\theta}_t|\boldsymbol{\theta}_s)$, where $p(\boldsymbol{\theta}_t|\boldsymbol{\theta}_s)$ is the prior distribution. To simplify the formulation, we approximate the integration of $p(\boldsymbol{\theta}_{t'})$ by the maximum a posterior (MAP) value of $\boldsymbol{\theta}_{t'}^*$. We obtain the MAP value by training the model $\boldsymbol{\theta}$ with inputs $\mathbf{x}_{t'}$ and pseudo labels $\mathbf{y}_{t'}$ starting from $\boldsymbol{\theta}_{s'}$. The formulation then is derived as:

$$\log p(\mathbf{y}_{t'}|\mathbf{x}_{t'}, \boldsymbol{\theta}_{s'}, \mathbf{X}_{t'}) = \log \int p(\mathbf{y}_{t'}|\mathbf{x}_{t'}, \boldsymbol{\theta}_{t'}^*)p(\hat{\mathbf{y}}_{t'}|\mathbf{x}_{t'}, \boldsymbol{\theta}_{s'}, \mathbf{X}_{t'})d\hat{\mathbf{y}}_{t'}. \tag{15}$$

To incorporate representative neighboring target information into the generation of $\hat{\mathbf{y}}_{t'}$, we further introduce the latent variable $\mathbf{w}_{t'}$ into eq. (15) and a variational posterior of the joint distribution $q(\hat{\mathbf{y}}_{t'}, \mathbf{w}_{t'})$. The formulation is then derived as:

$$\log p(\mathbf{y}_{t'}|\mathbf{x}_{t'}, \boldsymbol{\theta}_{s'}, \mathbf{X}_{t'})$$
$$= \log \int \int p(\mathbf{y}_{t'}|\mathbf{x}_{t'}, \boldsymbol{\theta}_{t'}^*)p(\hat{\mathbf{y}}_{t'}, \mathbf{w}_{t'}|\mathbf{x}_{t'}, \boldsymbol{\theta}_{s'}, \mathbf{X}_{t'})d\hat{\mathbf{y}}_{t'}d\mathbf{w}_{t'}$$
$$= \log \int \int p(\mathbf{y}_{t'}|\mathbf{x}_{t'}, \boldsymbol{\theta}_{t'}^*)\frac{p(\hat{\mathbf{y}}_{t'}, \mathbf{w}_{t'}|\mathbf{x}_{t'}, \boldsymbol{\theta}_{s'}, \mathbf{X}_{t'})}{q(\hat{\mathbf{y}}_{t'}, \mathbf{w}_{t'}|\mathbf{x}_{t'}, \boldsymbol{\theta}_{s'}, \mathbf{X}_{t'}, \mathbf{Y}_{t'})}q(\hat{\mathbf{y}}_{t'}, \mathbf{w}_{t'}|\mathbf{x}_{t'}, \boldsymbol{\theta}_{s'}, \mathbf{X}_{t'}, \mathbf{Y}_{t'})d\hat{\mathbf{y}}_{t'}d\mathbf{w}_{t'}, \tag{16}$$

where $p(\hat{\mathbf{y}}_{t'}, \mathbf{w}_{t'}|\mathbf{x}_{t'}, \boldsymbol{\theta}_{s'}, \mathbf{X}_{t'}) = p(\hat{\mathbf{y}}_{t'}|\mathbf{w}_{t'}, \mathbf{x}_{t'})p_\phi(\mathbf{w}_{t'}|\boldsymbol{\theta}_{s'}, \mathbf{X}_{t'})$ denote the prior distribution and $q(\hat{\mathbf{y}}_{t'}, \mathbf{w}_{t'}|\mathbf{x}_{t'}, \boldsymbol{\theta}_{s'}, \mathbf{X}_{t'}, \mathbf{Y}_{t'}) = p(\hat{\mathbf{y}}_{t'}|\mathbf{w}_{t'}, \mathbf{x}_{t'})q_\phi(\mathbf{w}_{t'}|\boldsymbol{\theta}_{s'}, \mathbf{X}_{t'}, \mathbf{Y}_{t'})$ denotes the posterior one. By incorporating the prior and posterior distribution into eq. (16), the formulation is derived to:

$$\log p(\mathbf{y}_{t'}|\mathbf{x}_{t'}, \boldsymbol{\theta}_{s'}, \mathbf{X}_{t'})$$
$$= \log \int \int p(\mathbf{y}_{t'}|\mathbf{x}_{t'}, \boldsymbol{\theta}_{t'}^*)\frac{p_\phi(\mathbf{w}_{t'}|\boldsymbol{\theta}_{s'}, \mathbf{X}_{t'})}{q_\phi(\mathbf{w}_{t'}|\boldsymbol{\theta}_{s'}, \mathbf{X}_{t'}, \mathbf{Y}_{t'})}p(\hat{\mathbf{y}}_{t'}|\mathbf{w}_{t'}, \mathbf{x}_{t'})q_\phi(\mathbf{w}_{t'}|\boldsymbol{\theta}_{s'}, \mathbf{X}_{t'}, \mathbf{Y}_{t'})d\hat{\mathbf{y}}_{t'}d\mathbf{w}_{t'}$$
$$\geq \int \int \log p(\mathbf{y}_{t'}|\mathbf{x}_{t'}, \boldsymbol{\theta}_{t'}^*)p(\hat{\mathbf{y}}_{t'}|\mathbf{w}_{t'}, \mathbf{x}_{t'})q_\phi(\mathbf{w}_{t'}|\boldsymbol{\theta}_{s'}, \mathbf{X}_{t'}, \mathbf{Y}_{t'})d\hat{\mathbf{y}}_{t'}d\mathbf{w}_{t'}$$
$$+ \int \log \frac{p_\phi(\mathbf{w}_{t'}|\boldsymbol{\theta}_{s'}, \mathbf{X}_{t'})}{q_\phi(\mathbf{w}_{t'}|\boldsymbol{\theta}_{s'}, \mathbf{X}_{t'}, \mathbf{Y}_{t'})}q_\phi(\mathbf{w}_{t'}|\boldsymbol{\theta}_{s'}, \mathbf{X}_{t'}, \mathbf{Y}_{t'})d\mathbf{w}_{t'}$$
$$= \mathbb{E}_{q_\phi(\mathbf{w}_{t'})}\mathbb{E}_{p(\hat{\mathbf{y}}_{t'}|\mathbf{w}_{t'}, \mathbf{x}_{t'})}[\log p(\mathbf{y}_{t'}|\mathbf{x}_{t'}, \boldsymbol{\theta}_{t'}^*)] - \mathbb{D}_{KL}[q_\phi(\mathbf{w}_{t'}|\boldsymbol{\theta}_{s'}, \mathbf{X}_{t'}, \mathbf{Y}_{t'})||p_\phi(\mathbf{w}_{t'}|\boldsymbol{\theta}_{s'}, \mathbf{X}_{t'})]. \tag{17}$$

## A.2   FURTHER ANALYSES OF THE PROPOSED METHOD ON TEST-TIME GENERALIZATION

The goal of test-time generalization is to optimize for the expected classification performance during test time, i.e., $p(\mathbf{y}_t|\mathbf{x}_t, \boldsymbol{\theta}_t)$. The model $\boldsymbol{\theta}_t$ is obtained by $p(\mathbf{x}_t)$ through $p(\boldsymbol{\theta}_t|\mathbf{x}_t, \boldsymbol{\theta}_s)$ in common test-time adaptation or test-time generalization methods, which is achieved by entropy minimization or pseudo-labeling based on the prediction on $\mathbf{x}_t$ of the source model $\boldsymbol{\theta}_s$. However, due to distribution shifts, the prediction of $\mathbf{x}_t$ by $\boldsymbol{\theta}_s$ can be overconfident and mispredicted.

By introducing probabilistic pseudo labels, we consider their uncertainty during adaptation, which mitigates overconfidence and introduces the discriminative information more reasonably. For example, consider a toy binary classification task in Figure 6, where the predicted probability is $[0.4, 0.6]$ with the ground-truth label $[1, 0]$. The pseudo label generated by selecting the maximum probability is

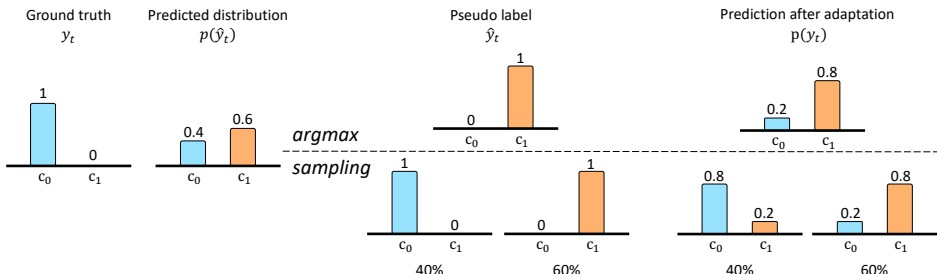

Figure 6: **Benefits of probabilistic pseudo labels.** Probabilistic pseudo labels consider the uncertainty to access correct supervision of the uncertain predictions, leading to more robust generalization than the common pseudo labels selected from the maximum probability.

$[0, 1]$, which is inaccurate. Optimization based on these labels would give rise to a model misspecified to target data, failing to generalize to the target domain. In contrast, our probabilistic formulation allows us to sample pseudo labels from the categorical distribution $p(\hat{\mathbf{y}}_t | \mathbf{x}_t, \boldsymbol{\theta}_s)$, which incorporates the uncertainty of the pseudo label $\hat{\mathbf{y}}_t$ in a principled way. Continuing the example, the pseudo labels sampled from the predicted distribution have a probability of $40\%$ to be the true label, which leads to the generalization of the model in the correct direction. Therefore, the formulation improves generalization by accessing accurate pseudo labels. The ablation studies in Table 1 (first two rows) and Figure 2 also demonstrate this.

We can also assume that the Oracle model for the target domain is obtained by $p(\boldsymbol{\theta}_t | \mathbf{x}_t, y_t, \boldsymbol{\theta}_s)$ while we lack the categorical information in $\mathbf{y}_t$. Theoretically, when the pseudo labels $p(\hat{\mathbf{y}}_t)$ carry richer and more accurate discriminatory information about $p(\mathbf{x}_t, \mathbf{y}_t)$, $\boldsymbol{\theta}_t$ adapts better to the target domain. Incorporating the neighboring information leverages category clusters from nearby target samples, enhancing target-specific cues in pseudo labels. Consequently, the model attained through variational neighbor labels yields more robust generalization of target data, which is also demonstrated in Table 1 (row 3). Moreover, with meta-learning, the model is trained to achieve good performance on target data with adaptation with variational pseudo labels, further improving the generalization ability. (Table 1 row 4).

## B ALGORITHMS

In the following section, we provide the algorithms for source training and test-time generalization in Algorithm 1 and 2, respectively.

## C IMPLEMENTATION DETAILS

Our training setup follows Iwasawa & Matsuo (2021), including the dataset splits and hyperparameter selection methods. The backbones such as ResNet-18 and ResNet-50 are pretrained on ImageNet same as the previous methods. As discussed in the main paper, the ERM baseline means we directly evaluate the source-trained model without any adjustment at test time (Gulrajani & Lopez-Paz, 2020). We also have other two baselines in the main paper. "Hard pseudo label" is obtained using the $argmax$ of model predictions and "soft pseudo label" refers to the original model predictions. Optimization with soft pseudo-labeling is similar to entropy minimization.

During training, we randomly select one source domain as the meta-target domain and the others as the meta-source domains in each iteration. The model is trained following the meta-source, meta-generalization, and meta-target stages as in Section 3 and Algorithm 1. We use a batch size of 70 for the model in the meta-training stage on source domains. To generate the variational neighbor labels, we implement the network '$\phi$' by a 3-layer MLP with 512 neurons per layer, which introduces approximately 1% more parameters than the backbone model in total. We use similar settings and hyperparameters for all domain generalization benchmarks that are reported in the main paper, i.e., 1e-4, 5e-5, and 1e-4 as $\lambda_1$, $\lambda_2$, and $\lambda_3$, respectively for Adam optimizer. The source-trained model

---

**Algorithm 1** Training for Meta-Generalized Variational Neighbor Labels

**Input:** $\mathcal{S} = \{D_s\}_{s=1}^{S}$: source domains with $n$ sample pairs $(\mathbf{x}_s, \mathbf{y}_s)$ for each; $\boldsymbol{\theta}$: model parameters of backbone; $\boldsymbol{\phi}$: model parameters of variational-neighbor-label generation; $\lambda_{1,2,3}$: learning rates; $\mathcal{B}_{tr}$: batch size during training; $N_{iter}$: the number of iterations.

**Output:** Learned $\boldsymbol{\theta}, \boldsymbol{\phi}$

---

1: **for** *iter* in $N_{iter}$ **do**
2:     $\mathcal{T}' \leftarrow$ Randomly Sample $(\{D_s\}_{s=1}^{S}, t')$;
       $\mathcal{S}' \leftarrow \{D_s\}_{s=1}^{S} \setminus \mathcal{T}'$;
3:     Sample datapoints $\{(\mathbf{x}_{s'}^{(k)}, \mathbf{y}_{s'}^{(k)})\}_{k=1}^{\mathcal{B}_{tr}} \sim \mathcal{S}'$, $\{(\mathbf{x}_{t'}^{(k)}, \mathbf{y}_{t'}^{(k)})\}_{k=1}^{\mathcal{B}_{tr}} \sim \mathcal{T}'$.
4:     **Meta-source stage:**
5:     Obtain meta-source model by training with the cross-entropy loss on meta-source labels and predictions
       $\boldsymbol{\theta}_{s'} = \min_{\boldsymbol{\theta}} \mathbb{E}_{(\mathbf{x}_{s'}, \mathbf{y}_{s'})) \in \mathcal{D}_{s'}}[L_{\mathrm{CE}}(\mathbf{x}_{s'}, \mathbf{y}_{s'}; \boldsymbol{\theta})]$.
6:     **Meta-generalization stage:**
7:     Generate the prior $p_{\boldsymbol{\phi}}(\mathbf{w}_{t'}|\boldsymbol{\theta}_{s'}, \mathbf{X}_{t'})$ and posterior distributions $q_{\boldsymbol{\phi}}(\mathbf{w}_{t'}|\boldsymbol{\theta}_{s'}, \mathbf{X}_{t'}, \mathbf{Y}_{t'})$ of $\mathbf{w}_{t'}$
8:     Sample the variational neighbor labels from the variational posterior distribution
       $\hat{\mathbf{y}}_{t'} \sim p(\hat{\mathbf{y}}_{t'}|\mathbf{w}_{t'}, \mathbf{x}_{t'})$,   $\mathbf{w}_{t'} \sim q_{\boldsymbol{\phi}}(\mathbf{w}_{t'}|\boldsymbol{\theta}_{s'}, \mathbf{X}_{t'}, \mathbf{Y}_{t'})$.
9:     Generalize the meta-source model to meta-target by cross-entropy loss with variational neighbor labels:
       $\boldsymbol{\theta}_{t'}^{*} = \boldsymbol{\theta}_{s'} - \lambda_1 \nabla_{\boldsymbol{\theta}} L_{\mathrm{CE}}(\mathbf{x}_{t'}, \hat{\mathbf{y}}_{t'}; \boldsymbol{\theta}_{s'})$.
10:    **Meta-target stage:**
11:    Calculate meta-target loss on the generalized meta-target model
       $\mathcal{L}_{meta} = \mathbb{E}_{(\mathbf{x}_{t'}, \mathbf{y}_{t'})}[\mathbb{E}_{q_{\boldsymbol{\phi}}(\mathbf{w}_{t'})} \mathbb{E}_{p(\hat{\mathbf{y}}_{t'}|\mathbf{w}_{t'}, \mathbf{x}_{t'})} L_{\mathrm{CE}}(\mathbf{x}_{t'}, \mathbf{y}_{t'}; \boldsymbol{\theta}_{t'}^{*})] + \mathbb{D}_{KL}[q_{\boldsymbol{\phi}}(\mathbf{w}_{t'})||p_{\boldsymbol{\phi}}(\mathbf{w}_{t'})]$.
12:    Calculate the cross-entropy loss on the variational neighbor labels and actual labels $\mathcal{L}_{\acute{c}e} = L_{CE}(\hat{\mathbf{y}}_{t'}, \mathbf{y}_{t'})$.

13:    Update the parameters $\boldsymbol{\theta}$ and $\boldsymbol{\phi}$ by $\boldsymbol{\theta} = \boldsymbol{\theta}_{s'} - \lambda_2 \nabla_{\boldsymbol{\theta}} \mathcal{L}_{meta}$ and $\boldsymbol{\phi} = \boldsymbol{\phi} - \lambda_3 (\nabla_{\boldsymbol{\phi}} \mathcal{L}_{\acute{c}e} - \nabla_{\boldsymbol{\phi}} \mathcal{L}_{meta})$,
       respectively. //Note the meta-target loss optimizes the meta-source model $\boldsymbol{\theta}_{s'}$.
14: **end for**

---

**Algorithm 2** Test-time Generalization by Meta-Generalized Variational Neighbor Labels

**Input:** $\mathcal{T}$: target domain with $N_t$ samples $\mathbf{x}_t$; $\boldsymbol{\theta}_s, \boldsymbol{\phi}_s$: source trained model parameters; $\lambda_1$: learning rate for test-time generalization; $\mathcal{B}_{te}$: batch size for each online step at test time.

---

1: Initialize $\boldsymbol{\theta}_t = \boldsymbol{\theta}_s$.
2: **for** *iter* in $(N_t/\mathcal{B}_{te})$ **do**
3:     Sample one batch of target samples from the target domain $\{(\mathbf{x}_t^{(k)})\}_{k=1}^{\mathcal{B}_{te}} \sim \mathcal{T}$.
4:     Sample the variational neighbor labels for the target batch from the prior distribution
       $\hat{\mathbf{y}}_t \sim p(\hat{\mathbf{y}}_t|\mathbf{w}_t, \mathbf{x}_t)$,   $\mathbf{w}_t \sim p_{\boldsymbol{\phi}}(\mathbf{w}_t|\boldsymbol{\theta}_t, \mathbf{X}_t)$.
5:     Generalize the model parameters by cross-entropy loss with variational neighbor labels:
       $\boldsymbol{\theta}_t^{*} = \boldsymbol{\theta}_t - \lambda_1 \nabla_{\boldsymbol{\theta}} L_{\mathrm{CE}}(\mathbf{x}_t, \hat{\mathbf{y}}_t; \boldsymbol{\theta}_t)$.
6:     Make predictions of the target current target batch by
       $p(\mathbf{y}_t|\mathbf{x}_t, \boldsymbol{\theta}_t, \mathbf{X}_t) = \mathbb{E}_{p_{\boldsymbol{\phi}_s}(\mathbf{w}_t)} \mathbb{E}_{p(\hat{\mathbf{y}}_t|\mathbf{w}_t, \mathbf{x}_t)}[\log p(\mathbf{y}_t|\mathbf{x}_t, \boldsymbol{\theta}_t^{*})]$.
7:     Update $\boldsymbol{\theta}_t = \boldsymbol{\theta}_t^{*}$
8: **end for**

---

with the highest train validation accuracy is selected as the initial model for test-time generalization on the target domain as in Iwasawa & Matsuo (2021).

At test time, we sample the variational neighbor labels directly from the prior distribution (eq. 5) and utilize the labels for updating the source-trained model. The source model is generalized to target data in an online manner with 20 target samples per batch, which is a small number. We update all parameters of the model with the learning rate of 1e-4 in an online manner using Adam optimizer. We choose the hyperparameters for model adjustment based on the validation set as mentioned in Gulrajani & Lopez-Paz (2020) and Iwasawa & Matsuo (2021). There are no additional hyperparameters involved in our method. We train and evaluate all our models on one NVIDIA Tesla 1080Ti GPU and utilize the PyTorch framework. We run all the experiments using 5 different random seeds and report the results. We will release the code.

**Additional dataset details.** We demonstrate the effectiveness of our method on image classification problems and evaluate it on seven widely used domain generalization datasets. Namely, *PACS* (Li et al., 2017) consists of 7 classes and 4 domains: Photo, Art painting, Cartoon, and Sketch with

Table 4: **Runtime required for source training on PACS using ResNet-18 as a backbone network.** The proposed method has overall larger runtime during training due to the meta-learning strategy but introduces few extra parameters.

|  | Parameters | Time for 10000 iterations |
|---|---|---|
| ERM baseline | 11.18 M | 6.5 hours |
| *VNL* | 11.96 M | 14.6 hours |

Table 5: **Runtime averaged for datasets using ResNet-18 as a backbone network.** The proposed method has similar or even better overall runtime at test time with the other test-time adaptation and test-time domain generalization methods.

|  | VLCS | PACS | Terra | OfficeHome |
|---|---|---|---|---|
| Wang et al. (2021) | 7m 28s | 3m 16s | 10m 34s | 7m 25s |
| Wang et al. (2021) | 2m 8s | 33s | 2m 58s | 1m 57s |
| Liang et al. (2020) | 8m 09s | 4m 22s | 12m 40s | 8m 38s |
| Jang et al. (2023) | 10m 34s | 9m 30s | 26m 14s | 22m 24s |
| Iwasawa & Matsuo (2021) | 2m 09s | 33s | 2m 59s | 2m 15s |
| *VNL* | 2m 20s | 5m 33s | 14m 30s | 7m 07s |

9,991 samples. *VLCS* (Fang et al., 2013) consists of 5 classes from 4 different datasets: Pascal, LabelMe, Caltech, and SUN with 10,729 samples. *Office-Home* (Venkateswara et al., 2017) contains 15,500 images of 65 categories and from four domains, i.e., Art, Clipart, Product, and Real-World. *TerraIncognita* (Beery et al., 2018) has 4 domains and 4 locations: L100, L38, L43, and L46. The dataset includes 24,778 samples of 10 categories. *Mini DomainNet* (Zhou et al., 2021) is a subset of DomainNet (Peng et al., 2019) with 140,000 samples with 4 domains and 126 classes. We follow training and validation split in (Li et al., 2017) and evaluate model according to "leave-one-out" protocol (Li et al., 2019; Carlucci et al., 2019). We also evaluate our method on the *Rotated MNIST* and *Fashion-MNIST* datasets following Piratla et al. (2020), where the images are rotated by different angles as different domains. We use subsets with rotation angles from $15°$ to $75°$ in intervals of $15°$ as five source domains, and images rotated by $0°$ and $90°$ as target domains.

## D  COMPUTATIONAL COMPLEXITY AND RUNTIME COMPARISON

We provide the overall runtime comparison of our method in both the training (Table 4) and test-time stage (Table 5). As we utilize the meta-learning-based strategy to learn the ability to handle domain shifts during training and to generate variational neighbor labels, the runtime during training is larger than the ERM baseline. Moreover, compared with the ERM baseline, our variational neighbor-labeling and meta-learning framework only introduces a few more parameters (around 1%).

Since the meta-learning strategy is only deployed in the training time, our method has similar overall runtime for generalization at test time compared with the other test-time adaptation, e.g., Tent (Wang et al., 2021), and test-time domain generalization methods, e.g., TAST (Jang et al., 2023). Another factor for the runtime is the number of updated parameters at test time. The models only update the parameters of BN layers, e.g., Tent-BN (Wang et al., 2021), or classifiers, e.g., T3A (Iwasawa & Matsuo, 2021) have lower overall runtime than ours that update all model parameters. Compared with the methods that update all parameters with pseudo labels (Liang et al., 2020; Jang et al., 2023), our overall computational runtime is competitive and even lower.

## E  ADDITIONAL RESULTS AND DISCUSSIONS

**Importance of the model adjustment at test time.**    We also provide extra ablation studies on the pseudo-label generation and generalization with our variational neighbor labels. We directly make predictions using the variational neighbor labels that are generated by sampling from the pseudo label distributions at test time. As shown in Table 6, the predictions based on the variational neighbor labels distributions are better than the ERM baseline, demonstrating that our variational neighbor labels are better than the original prediction of the source model, i.e., the common pseudo labels. Moreover,

after online adjusting the model parameters by our variational neighbor labels, the performance further improves, demonstrating the effectiveness of the model adjustment at test time.

Table 6: **Investigation of the model adjustment at test time.** We conduct ablation studies on PACS using ResNet-18. Our method that updates the source-trained model with variational neighbor labels performs better than the prediction directly sampled from the variational neighbor distributions.

| Settings | Photo | Art-painting | Cartoon | Sketch | *Mean* |
|---|---|---|---|---|---|
| ERM baseline | 94.10 ±0.4 | 78.00 ±1.3 | 73.40 ±0.8 | 73.60 ±2.2 | 79.80 ±0.4 |
| Prediction by pseudo label distributions | 95.97 ±1.0 | 81.23 ±0.5 | 79.19 ±0.7 | 73.22 ±0.5 | 82.40 ±0.7 |
| *VNL* | **96.40** ±0.2 | **83.81** ±0.4 | **82.62** ±0.5 | **77.20** ±0.7 | **85.00** ±0.4 |

**Additional comparisons on other datasets** We also conduct experiments on rotated MNIST, rotated Fashion-MNIST, and mini-DomainNet to provide another comparison as shown in Table 7 and Table 8. For rotated MNIST and Fashion-MNIST, we follow the settings in Piratla et al. (2020) and use ResNet-18 as the backbone and for mini-DomainNet we use ResNet-18 as the backbone. The conclusion is similar to the other datasets, we are at least competitive and sometimes better than alternative test-time adaptation and generalization methods.

**Performance on single source image corruption datasets.** Apart from existing domain generalization datasets, we also conducted experiments on the CIFAR-10-C dataset. We train on the original data and evaluate the model on 15 types of corruption similar to Tent (Wang et al., 2021). We achieved 21.60% as the error rate. In comparison, the source model without adaptation achieves 29.14 %, and Tent achieves 14.30 %. The performance is not good enough since we cannot mimic distribution shifts by meta-generalization in this single-source setting. We consider single-source domain generalization at test time a worthwhile avenue for future work, as highlighted in the conclusion.

**Large number of categories and model capacity.** We have experimented with different numbers of layers in the MLP for Office-Home by utilizing ResNet-18 as the backbone. We obtain a mean accuracy of 57.1 with 2 layers and 64.3 with 3 layers in $\phi$.

**Comparisons with additional existing domain generalization algorithms.** We provide additional domain generalization comparisons from Gulrajani & Lopez-Paz (2020) in the following Table 9 based on ResNet-50. Our method achieves better results compared with these methods.

**Comparisons to additional Test-time adaptation methods with pre-training phase.** Compared with the fully test-time adaptation methods (e.g., Tent (Wang et al., 2021)), our method modifies the learning strategy during source training. To further show the effectiveness of our method, we also compare our method with some test-time training methods, TTT (Sun et al., 2020) and TTT+ (Liu et al., 2021b), who also modify the learning strategy during training. We re-implemented their methods on the PACS dataset with ResNet-18 for test-time domain generalization. TTT (Sun et al., 2020) obtains 83.0% accuracy with ±0.2% standard deviation and Liu et al. (2021b) obtains 83.8 accuracy with ±0.5% standard deviation. Our method obtains 85.0 % accuracy with ±0.4% standard deviation and outperforms these methods.

## F  DETAILED EXPERIMENTAL RESULTS

**Detailed results of the ablation study on variational neighbor labels.** We provide the detailed results of Table 1 on PACS in Table 10. The conclusion is similar to that of Table 3. The probabilistic formulation of pseudo-labeling improves the performance of the common one on most of the domains for both PACS and TerraIncognita. Moreover, based on the probabilistic formulation, the variational neighbor labels incorporate more target information, which also improves results on most domains. Learned by the proposed meta-generalization, the variational neighbor labels further improve the performance on most domains.

**Detailed results of different batch sizes at test time.** Test-time generalization and adaptation methods usually require updating the source-trained model with large online batches of target samples, which is not always available in real-world applications. To show the robustness of our method on

Table 7: **Comparison on rotated MNIST and Fashion-MNIST.** The models are evaluated on the test sets of MNIST and Fashion-MNIST with rotation angles of $0°$ and $90°$. Again, our method achieves the best performance (bold) compared to state-of-the-art alternatives.

|  | MNIST | Fashion-MNIST |
|---|---|---|
| ERM Baseline | 93.5 | 76.9 |
| Wang et al. (2021) | 95.3 | 78.9 |
| Xiao et al. (2022) | **95.8** | 80.8 |
| *VNL* | **95.9** ±0.1 | **82.4** ±0.2 |

Table 8: **Comparison on mini-DomainNet.** Our method performs better (bold) compared to state-of-the-art alternatives.

|  | mini-DomainNet |
|---|---|
| ERM baseline | 60.5 |
| Li et al. (2018a) | 61.1 |
| Blanchard et al. (2021) | 62.1 |
| *VNL* | **63.1** ±0.3 |

limited data, we conduct experiments with different batch sizes during test-time generalization and compare the proposed method with Tent (Wang et al., 2021) in Figure 3 in the main paper. The experiments are conducted on PACS with ResNet-18. Here we provide detailed results of each domain in Table 11. The conclusion is similar to that in the main paper. Tent performs well with large batch sizes, e.g., 128, but fails with small ones, e.g., 16. The performance with batch sizes of 16 is even lower than the baseline model. By contrast, our method performs consistently better than Tent. Moreover, by incorporating the uncertainty and representative neighboring information, our method is more robust to small batch sizes, leading to a larger improvement than Tent.

**Detailed results of the method with augmentation at test time.** As discussed in the main paper, our variational neighbor labeling is orthogonal to other deployment techniques for test-time domain generalization, e.g., data augmentation at test time. We provide detailed results in Table 12 where we compare the results of the proposed method on PACS with and without augmenting target samples during generalization at test time. Our method achieves better results on all domains in conjunction with augmentations.

Table 9: **Comparisons to additional existing domain generalization algorithms** for ResNet-50 backbones on different datasets. Our method performs better (bold) than the other methods.

|  | PACS | VLCS | Office-Home | Terra Incognita |
|---|---|---|---|---|
| ERM baseline | 85.7 | 77.4 | 67.5 | 47.2 |
| Arjovsky et al. (2019) | 83.5 | 78.5 | 64.3 | 47.6 |
| Sagawa et al. (2019) | 84.4 | 76.7 | 66.0 | 43.2 |
| Li et al. (2018a) | 84.9 | 77.2 | 66.8 | 47.7 |
| Sun & Saenko (2016) | 86.2 | **78.8** | **68.7** | 47.6 |
| Li et al. (2018b) | 84.6 | 77.5 | 66.3 | 42.2 |
| Ganin et al. (2016) | 83.6 | 78.6 | 65.9 | 46.7 |
| Li et al. (2018d) | 82.6 | 77.5 | 65.8 | 45.8 |
| *VNL* | **87.9** ±0.3 | **79.1** ±0.4 | **69.1** ±0.4 | **49.4** ±0.6 |

Table 10: **Detailed ablations of the variational neighbor labels.** The experiments are conducted on PACS with ResNet-18. Our probabilistic formulation performs better than the common pseudo-labeling baseline on most of the domains by considering the uncertainty. Based on the probabilistic formulation, the variational neighbor labels incorporate more target information, which further improves results on most domains, especially learned by the proposed meta-generalization.

|  |  | Photo | Art-painting | Cartoon | Sketch | *Mean* |
|---|---|---|---|---|---|---|
| Pseudo-labeling baseline | (eq. 1) | 93.91 $\pm$0.2 | 78.52 $\pm$0.4 | 78.33 $\pm$0.5 | 74.37 $\pm$0.8 | 81.3 $\pm$0.3 |
| Probabilistic pseudo-labeling | (eq. 3) | 94.55 $\pm$0.2 | 80.07 $\pm$0.3 | 79.14 $\pm$0.6 | 74.29 $\pm$0.6 | 82.0 $\pm$0.2 |
| Variational neighbor-labeling | (eq. 4) | 96.12 $\pm$0.3 | 80.40 $\pm$0.5 | 80.09 $\pm$0.4 | 73.33 $\pm$0.7 | 82.4 $\pm$0.6 |
| Meta-generalization with variational neighbor labels | (eq. 10) | 95.50 $\pm$0.2 | 82.90 $\pm$0.4 | 81.28 $\pm$0.4 | 74.11 $\pm$0.6 | 83.5 $\pm$0.4 |

Table 11: **Detailed comparisons of Tent on PACS with different batch sizes.** Our method consistently performs better than Tent (Wang et al., 2021) with different batch sizes during test-time generalization. Moreover, the proposed method is more robust with small batch sizes.

|  | Photo | Art | Cartoon | Sketch | *Mean* |
|---|---|---|---|---|---|
| Baseline model | 92.40 $\pm$0.2 | 78.70 $\pm$1.3 | 74.30 $\pm$0.6 | 75.60 $\pm$0.8 | 80.30 $\pm$0.4 |
| **Generalization with 16 samples per step** | | | | | |
| Tent | 93.65 $\pm$0.3 | 80.20 $\pm$0.2 | 76.90 $\pm$0.5 | 68.49 $\pm$0.7 | 79.81 $\pm$0.3 |
| *VNL* | 95.45 $\pm$0.2 | 83.70 $\pm$0.4 | 80.42 $\pm$0.5 | 74.91 $\pm$0.7 | 83.62 $\pm$0.5 |
| **Generalization with 64 samples per step** | | | | | |
| Tent | 96.04 $\pm$0.3 | 81.91 $\pm$0.4 | 80.81 $\pm$0.6 | 76.33 $\pm$0.7 | 83.77 $\pm$0.4 |
| *VNL* | 96.09 $\pm$0.2 | 84.37 $\pm$0.4 | 81.25 $\pm$0.5 | 78.12 $\pm$0.7 | 85.00 $\pm$0.5 |
| **Generalization with 128 samples per step** | | | | | |
| Tent | 97.25 $\pm$0.2 | 84.91 $\pm$0.3 | 81.12 $\pm$0.5 | 76.80 $\pm$0.8 | 85.02 $\pm$0.5 |
| *VNL* | 96.75 $\pm$0.2 | 85.15 $\pm$0.4 | 85.70 $\pm$0.4 | 78.90 $\pm$0.6 | 86.62 $\pm$0.4 |

Table 12: **Variational neighbor labels combined with augmentation at test time on PACS.** Our method achieves better results in conjunction with augmentations.

| Data augmentation | Photo | Art | Cartoon | Sketch | *Mean* |
|---|---|---|---|---|---|
| ✗ | 95.50 $\pm$0.4 | 82.90 $\pm$0.3 | 81.28 $\pm$0. | 74.11 $\pm$0.6 | 83.45 $\pm$0.4 |
| ✓ | **96.40** $\pm$0.2 | **83.81** $\pm$0.4 | **82.62** $\pm$0.4 | **77.20** $\pm$0.6 | **85.00** $\pm$0.4 |

