# OpenReview forum: "Learning Variational Neighbor Labels for Test-Time Domain Generalization"
_ICLR.cc/2024/Conference — Submitted to ICLR 2024_

### Official Review · Reviewer_4tVC · 2023-10-26

**Soundness:** 2 fair
**Presentation:** 1 poor
**Contribution:** 2 fair
**Rating:** 5
**Confidence:** 4

**Summary:**

This paper considered test domain adaptation, where this paper considered variational latent labels (through a latent variable w) to better estimate pseudo-labels in the test time. Through a variational objective and meta-learning framework, different variables such as latent auxiliary variable (w_t) and target pseudo-labels (\hat_y_t) are estimated.
Finally, the model is deployed in standard benchmarks with improved performance.

========Post rebuttal

Thanks for the rebuttal. I would think paper still needs major revisions to improve the clarity.

**Strengths:**

This paper considered a reasonable solution in test time domain generalization. Through better estimating pseudo-labels, this paper obtained better results in different benchmark dataset.

**Weaknesses:**

The main issue in this paper is the **clarity** part. This paper considered probabilistic method and variational inference. Many parts are not clear or seemingly not correct in my viewpoint. I do think a major revision is required for the resubmission.

1. As for the graphical model in Fig 1, I feel quite confused at first glance. In fact, it is not a real probabilistic graphical model for the data, but rather a model/data interaction graph. I think this should be clarified for the potential misunderstanding.

2.  Could authors provide detailed explanations on the followings:

  (1) How to estimate the following conditional probabilities?
-  P_\phi (w_t | X_t, \theta_s)
- Why is a data batch written as a random variable X_t?
- q_\phi (w_t | X_t, Y_t \theta_s)
- We never observed Y_t, right ?

(2)  What is the rationale of meta-generalization? What is the corresponding graph?

(3) If a model parameter is updated through gradient descent like equation (11), it should not be considered as a formal variational inference. Indeed, the gradient flow could make it hard to construct a probabilistic term.

3. I still could not understand the specific reason to use variational inference based methods. I think many math equations could be replaced by deterministic updates and significantly simplified.
4. The quality of pseudo-label is still unknown and hard to understand. Why could such a method improve the pseudo quality? When will it happen?

**Questions:**

See the weakness part.

---

> ### Author Response · Authors · 2023-11-22
> **Response to reviewer 4tVC**
>
> We thank Reviewer 4tVC for their feedback.
>
>
> **Graphical model**
>
> We agree with the reviewer that the graphical model might be misleading. We meant to express the conditional dependence structure between random variables (data and latent variables). We changed it to “Probabilistic modeling graph” in the updated paper.
>
> **Estimation conditional probabilities**
>
> (1) For the prior distribution $p_\phi (w_t | X_t, \theta_s)$, we first generate the features $z_t$ of the batch of images $X_t$ and their pseudo labels $\hat{Y}_{t}$ through the source trained model $\theta_s$.
>
> Then we generate the averaged features $z_t^c$ for each category according to the pseudo labels. The categorical averaged features $z_t^c$ are then fed to the MLP network $\phi$ to generate the mean and variance of $p_\phi(w_t)$. $X_t$ here is not a random variable. It denotes the batch of images, similar to the image $x_t$.
>
> (2) For the posterior distribution $q_\phi (w_t | X_t, Y_t, \theta_s)$, the reviewer is right that we never observe $Y_t$, so the distribution is intractable at test time. Therefore we use only the prior distribution for the test and introduce meta-generalization to estimate the posterior distribution for the meta-target data during training. The meta-target domain is simulated by the multiple source domains, so we can get $Y_{t’}$ during training.
> Similar to the prior distribution, we first generate the features $z_{t’}$ of the batch of images $X_{t’}$. Differently, the categorical averaged features are generated according to the ground truth meta-target labels $Y_{t’}$. Then we use the same MLP $\phi$ to generate q(w_t) by the categorical averaged features.
>
>
>
> **Rationale of meta-generalization and graph**
>
> Meta-generalization is utilized to simulate distribution shifts during training by splitting the multiple source domains into meta-source and meta-target domains. By doing so, we learn to generalize over shifts between meta-source and meta-target distributions. At test time, we leverage this learned ability to generalize over the unseen target data. The meta-generalization mechanism alleviates the overfitting of directly learning a model on source data.
>
> Under meta-generalization, the graph can be directly obtained by replacing the source and target variables in Figure 1 (c) with meta-source and meta-target variables.
>
> **Gradient descent makes it hard to construct a probabilistic term**
>
> We agree with the reviewer that gradient descent makes it hard to construct a probabilistic term. Since the distribution of the model parameters is unknown, we have to rely on an empirical Bayesian method with gradient descent to approximate $\theta_t$ (Chelsea Finn et al., NeurIPS 2018).
>
> Note we only use this approximation to estimate the final model parameter $\theta_t$, the variational inference is used to generate the pseudo labels. During the generation of the variational neighbor labels, we never use gradient descent to approximate the probabilistic term.
>
> We have clarified this in the methodology section.
>
> **Reason to use variational inference**
>
> Our key idea is to capture the uncertainty of the pseudo labels in the latent space for more robust generalization across distribution shifts. The probabilistic framework encodes the uncertainty and more target information for model generalization (e.g., the sample in Figure 6 in the appendix).
>
> To optimize the probabilistic framework, we use variational inference to approximate the true posterior of the probabilistic pseudo labels, which inherently introduces more neighboring target information and categorical information during training.
> Our experimental results in Table 1 (as well as the additional ablations by Reviewer Hsmr) confirm the variational pseudo labels outperform the deterministic ones, especially when combined with our proposed meta-generalization.
>
> We have added these discussions to the main paper.
>
>
>
> **Why does the method improve the pseudo label quality and when**
>
> By meta-generalization we simulate distribution shifts during training, so the model learns to generate more effective pseudo labels for fine-tuning the model across distribution shifts. The variational neighbor labels are further improved by considering more neighboring target information.
>
> It works because we have multiple source domains available during training to mimic distribution shifts and the availability of a batch of target samples for neighboring information at test time.
>
> We have added this discussion to the methodology section of the paper.

---

### Official Review · Reviewer_CA6h · 2023-10-29

**Soundness:** 3 good
**Presentation:** 2 fair
**Contribution:** 2 fair
**Rating:** 5
**Confidence:** 3

**Summary:**

In the presented paper, the authors advocate for domain generalization by developing models trained meticulously on source domains and subsequently deploying them on unexplored target domains. The authors unfold their contributions through strategies like probabilistic pseudo-labeling and meta-generalization stages, which seem instrumental in optimizing the performance of source-trained models when introduced to target domains.

The paper claims superior algorithmic performance in comparison to preceding methodologies. Experiments conducted seem to underpin the effectiveness of the proposed strategies, attesting to their potential relevance in domain generalization.

Despite its merits, the paper's exposition of integrating uncertainty with meta-generalized neighborhood information appears somewhat ambiguous. While leveraging neighborhood information for generalization has been a commonplace strategy in prior studies, the paper could benefit from a clearer elucidation of the novelty in their approach, particularly concerning the integration of uncertainty.

Furthermore, the validation scope of the proposed model seems narrowly focused, lacking a diverse spectrum of benchmarks for thorough evaluation. Inclusion of broader benchmarks could enhance the rigor of the validation process, presenting a more holistic view of the model’s adaptability and performance across varied scenarios. Such an extended validation would embolden the research’s integrity, offering a more comprehensive insight into its applicability and effectiveness.

**Strengths:**

- **Superior Performance:** The authors have claimed that the proposed algorithm outperforms previous methods, showcasing its effectiveness and superiority in achieving enhanced results in the conducted experiments and evaluations.

**Weaknesses:**

- **Ambiguity in Contribution regarding Meta-generalization and Uncertainty Utilizing Neighborhood Information:** Table 1 suggests that meta-generalization is the key step. The utilization of neighborhood information with uncertainty as a tactic for generalization is not novel. Such strategies have been previously explored and applied in various studies. The paper should explicitly articulate the unique contributions made in terms of incorporating uncertainty with meta-generalization. Delineating how uncertainty with meta-generalization has been applied or integrated in this study as a key contribution is essential for understanding the paper's novelty and significance.

- **Limited Validation:** The paper lacks extensive validation across a diverse range of benchmarks. Including additional benchmarks such as **STL10, CIFAR10-C, or CIFAR100-C, etc..** at least would strengthen the evaluation process and enhance the generalizability and applicability of the proposed model. **For this exercise, I suggest not to utilize other corrupted domains for CIFAR10-C or CIFAR100-C, then, it's a good validation of the proposed algorithm with single source domain input with unseen target samples.** i.e., $S=1$ in Algorithm 1. For this validation, it doesn't have to show superior performance, but such an expansion in validation datasets would provide a more comprehensive and rigorous assessment of the model's performance and robustness in various scenarios, helping to establish its efficacy and reliability more convincingly.

**Questions:**

- **Regarding meta-generalization training, does the process utilize information from the target domain "training" samples?** This critical question arises due to a statement on page 4, page 5, and Algorithm 1 mentioning the accessibility of actual labels of the meta-target data during training. I need confirmation and understanding this aspect is essential for assessing the generalizability and **the source of superior performance** of the proposed algorithm.

- What advantages does meta-generalization offer when compared to a general domain adaptation, including source-free domain adaptation? For example, SHOT (Liang et al. 2020) is a source-free domain adaptation. The meta-generalization steps require the leverage of various source domains. Could the authors elucidate the unique benefits that meta-generalization brings to enhance the model's performance or adaptability in domain generalization tasks? This clarification would help in understanding the specific improvements or innovations that meta-generalization contributes beyond the capabilities of existing source-free domain adaptation approaches. **I suggest providing a more detailed and concrete discussion of the benefits of the proposed method beyond what is explained in Section 4.** For example, it's better to include the ablation study regarding hyperparameters of $\lambda_1,\lambda_2,\lambda_3$ indicating the contribution of each factor.

- For the update of the variational neighbor label generator, could the authors clarify the rationale behind employing the difference of two losses in $L_{CE}-\mathcal{L}_{meta}$? I am curious about the motivation for the negative sign of meta loss.

- Could the authors elaborate on why there is a **noticeable performance degradation in the Office-Home dataset** as shown in Table 3, especially when compared to the results reported by Xiao et al. (2022)? An explanation regarding this discrepancy would be helpful for a more comprehensive understanding of the algorithm’s performance across different datasets.

- Algorithm 1 indicates the use of $n$ samples for each domain. Could the authors provide guidance on how to effectively balance these samples across various domains to ensure a harmonized and representative dataset for each domain involved?

- Are there any other hyper-parameters, aside from the learning rates $\lambda_1, \lambda_2, \lambda_3$, that are crucial in the model's implementation and performance?

- For further validation of the proposed algorithm, consider including **exercises in single source domain generalization**. Utilizing additional benchmarks, such as STL10, CIFAR10-C, or CIFAR100-C, would not only strengthen the evaluation process but also enhance the model's **generalizability** and applicability. Specifically, applying the algorithm in scenarios with a single source domain input and unseen target samples could provide a comprehensive validation of its effectiveness and robustness. For example, if the target domain is highly distorted (like some of the corruptions in CIFAR10/100-C), the selected hyperparameters might not suit the (unseen) target domain.

- **The supplementary code provided appears to be non-executable in its current form**, and it lacks essential implementations pivotal to this concept. For example, in "dg_adapt_sampler.py", it seems that there are missing/or modified segments of code crucial for execution. I could find a lot more missing pieces, so it's impossible to validate the key implementation. To facilitate a better understanding and application of the idea presented, please consider supplying a more comprehensive and runnable version of the code that includes all key implementations and necessary details for successful execution.

---

> ### Author Response · Authors · 2023-11-22
> **Response to reviewer CA6h**
>
> We thank Reviewer C6Ah for their feedback.
>
> **Ambiguity in Contribution**
>
> We agree that meta-generalization is important and helps us achieve a good improvement. By simulation of distribution shifts during training the model learns the ability to better encode uncertainty in the variational neighbor labels and improve generalization across distribution shifts, see also the requested ablation by Reviewer Hsmr. We will better stress its importance in the updated paper.
>
> **Limited validation**
>
> We perform experiments on seven datasets that cater for (test-time) domain generalization, as we require multiple domains for training (similar to Gulrajani & Lopez-Paz, 2020; Iwasawa & Matsuo., 2021; Xiao et al., 2022). The suggested datasets are all single-source and used for (test-time) robustness to image corruptions. Nevertheless, we conducted the experiment on Cifar-10-C and obtained 21.60 as the error rate, where the source model without adaptation achieves 29.14 and Tent (Wang et al., 2021) obtains 13.95. We have added this comparison in the appendix. The dependence on multiple source domains during training is both a strength and a weakness of our proposal. We consider single-source domain generalization at test time a worthwhile avenue for future work, as highlighted in the conclusion.
>
> **Regarding utilizing information from the target domain training samples**
> We would like to stress that we do not utilize any target samples during training. Instead, we mimic domain shifts during source training by obtaining a “meta target” set from the source domains. Specifically, during source training, we use one source domain as the meta-source and the other source domains as the meta-target in each iteration (this is the reason why we depend on datasets providing multiple domains).
>
> **Benefits of meta-generalization**
>
> The benefit of meta-generalization is to simulate distribution shifts during training. By doing so, the model learns the ability of generalization across domain shifts. At test time, the method can use the learned generalization ability for the (unseen) distribution shift between source and target data, leading to more robust generalization compared with methods without meta-learning.
> To further show the benefits of mimicking distribution shifts by meta-generalization. We conduct more ablations based on meta-generalization. As in the answer of Reviewer Hsmr, meta-generalization can also improve the performance of common pseudo-labeling and probabilistic pseudo-labeling, showing the effectiveness of mimicking distribution shifts. Meta-generalization of variational neighboring labels gains more improvement and the best overall results.
>
> We also considered meta-generalization with fewer distribution shifts during training. When we train on two instead of three source domains from PACS we obtain 80.32±1.8 on the “art-painting” target set, where using all three domains gives 83.81. Showing the importance of mimicking distribution shifts for meta-generalization.
>
> **Difference between \L_{ce} and \L_{meta}**
>
> The negative sign is a typo, it should be a positive sign. We have updated it accordingly. $L_{ce}$ calculates the cross-entropy between neighbor labels and the ground truth, while $L_{meta}$ calculates the cross-entropy between the updated model's prediction and ground truth.
>
> **Performance degradation on Office-Home**
>
> The reviewer is correct. We attribute the performance degradation for Office-Home to the larger number of 65 categories, demanding a larger model capacity $\phi$. We have varied $\phi$ with different numbers of layers in the MLP for Office-Home. We obtain a mean accuracy of 57.1 with 2 layers and 64.3 with 3 layers. We have discussed this in the main paper and kept the details of the experiment in the appendix. Thank you.
>
> **Clarification about n samples in Algorithm 1**
>
> We just use the numbers “n” provided by the public datasets. Our method is not sensitive to “n” since we randomly select the meta-source and meta-target samples in each iteration.
>
>
> **Additional hyperparameters**
>
> Similar to recent methods, we have ablated standard test-time hyperparameters such as batch size. Additionally, since we use a small $\phi$ network we have also ablated its model capacity and added it in the main paper. There are no additional hyperparameters involved. We have clarified this in the main paper.
>
> **Code**
> We added more implementation details and updated the code, and will include a link to the public github repository in the final version.

---

> > ### Comment · Reviewer_CA6h · 2023-11-23
> >
> > I am thankful to the authors for taking the time to address my questions and concerns. In light of their responses, I maintain my initial score.

---

### Official Review · Reviewer_JCc1 · 2023-11-01

**Soundness:** 2 fair
**Presentation:** 2 fair
**Contribution:** 2 fair
**Rating:** 5
**Confidence:** 3

**Summary:**

In this work, the authors propose a test-time domain generation method from a probabilistic perspective by modeling pseudo labels as distributions. Specifically, variational neighbor labels are incorporated to generate more robust pseudo labels, and meta-learning-based algorithms are proposed to boost the performance. The experiments on seven datasets show its effectiveness.

**Strengths:**

+ The experiments conducted on seven datasets show its superiority over SOTA, with subtle improvement.
+ code is provided for reproducibility.

**Weaknesses:**

- The experiments are limited to small-scale datasets.  Some commonly adopted large-scale datasets are missing, e.g. CIFAR10-C, CIFAR100-C, ImageNet-C, VisDA.
- The amount of improvement is limited. Ablation study on variational neighbor labels shows a subtle improvement from neighbor-labeling.
- Presentation:
   - The paper writing needs improvement, which is hard to follow.
   - I suggest labeling the proposed approach with a meaningful code instead of "this paper".
   - It is better to move section 4, "related work" ahead the following introduction.
- The hyperparameter selection, e.g. learning rate,  for experiments on every dataset are not explained.
-  what is the inference speed. How much computation cost does this neighbor pseudo label introduce?

**Questions:**

please check the weakness section.

---

> ### Author Response · Authors · 2023-11-22
> **Response to reviewer JCc1**
>
> We thank Reviewer JCc1 for their feedback.
>
>
>
> **Scale of datasets**
>
> We agree with the reviewer that the commonly used datasets PACS (9,991 images with 7 classes) and VLCS (10,729 images with 5 classes) are relatively small-scale. That is why we also conducted experiments on the larger-scale datasets Office-Home (15,588 images with 65 classes) and mini-DomainNet (140,000 images with 126 classes) in the appendix. On all datasets our method achieves competitive performance.
>
> Since we utilize meta-learning to mimic domain shifts and neighboring target information, we indeed require multiple source domains (Iwasawa & Matsuo., 2021) during training as supported by the domain generalization datasets PACS, VLCS, Terra Incognita, Office-Home and mini-DomainNet. Consequently, it is difficult to conduct our method on single-source datasets intended for image corruption robustness, like CIFAR-10C (70,000 images with 10 classes) and ImageNet-C (14,197,122 images with 1000 classes). Nevertheless, we conducted the experiment on Cifar-10-C and obtained 21.60 as the error rate, where the source model without adaptation achieves 29.14 and Tent (Wang et al., 2021) achieves 14.30. We have added this comparison in the appendix. The dependence on multiple source domains during training is both a strength and a weakness of our proposal. We consider single-source domain generalization at test time a worthwhile avenue for future work, as highlighted in the conclusion.
>
>
>
> **Improvement over prior work**
>
> We achieve state-of-the-art on 6 of 7 domain generalization datasets, and the improvement per dataset varies from 0.4% (VLCS) to 2.1% (Terra Incognita). We observe our peers report similar improvement gains (e.g., Iwasawa & Matsuo, 2022, Chen et al., 2023b, Jang et al., 2023, Xiao et al., 2022), while achieving state-of-the-art on fewer datasets (e.g., Iwasawa & Matsuo, 2022 on 2 out of 4 datasets reported; Chen et al., 2023 on 3 out of 5 datasets reported). Alongside its empirical performance, our method also achieves better calibration (Figure 2) and addresses complex scenarios (Figure 5).
>
>
> **Presentation**
>
> Thank you for the suggestions. We have relocated related work and used the more meaningful codename ‘VNL’ for our method in the updated paper.
>
>
>
>
> **Other datasets hyperparameters**
>
> We use similar settings and hyperparameters for all domain generalization benchmarks that are reported in the main paper. We have moved the description from the appendix to the main paper.
>
> **Inference speed and computational cost**
>
> Although we are not as fast at inference as the online classifier adjustment method by Iwasawa & Matsuo., (2021) and BN fine-tuning methods (Wang et al., 2021 (BN)), we are comparable to other test-time domain generalization methods (Jang et al., 2023, Liang et al., 2020) as shown in the following table.
>
>
>
> |                            | VLCS   | PACS   | Terra   | OfficeHome |
> |----------------------------|--------|--------|---------|------------|
> | Wang et al., 2021          | 7m 28s | 3m 16s | 10m 34s | 7m 25s     |
> | Wang et al., 2021 (BN)     | 2m 8s  | 33s    | 2m 58s  | 1m 57s     |
> | Liang et al., 2020         | 8m 09s | 4m 22s | 12m 40s | 8m 38s     |
> | Jang et al., 2023          | 10m 34s| 9m 30s | 26m 14s | 22m 24s    |
> | Iwasawa & Matsuo., 2021    | 2m 09s | 33s    | 2m 59s  | 2m 15s     |
> | Variational Neighbor labels| 2m 20s | 5m 33s | 14m 30s | 7m 07s     |
>
>
> We introduce about 1% more parameters than the backbone model, while keeping computational costs low. We have mentioned the result of this experiment and the computational cost in the main paper and kept the Table in the appendix.

---

### Official Review · Reviewer_Hsmr · 2023-11-03

**Soundness:** 3 good
**Presentation:** 3 good
**Contribution:** 3 good
**Rating:** 5
**Confidence:** 3

**Summary:**

This paper presents three novel contributions aimed at addressing the issue of unreliable test-time domain generalization using pseudo labels. The authors' first contribution involves defining pseudo labels as stochastic variables and estimating their distributions, enabling the modeling of uncertainty in the predictions obtained from the source-trained models. Secondly, the authors propose the learning of variational neighbor labels to enhance the robustness of the pseudo labels. Lastly, they introduce a meta-generalization method that allows for the learning of variational neighbor labels during training, enabling the models to adapt to domain shifts. The authors support their claims with comprehensive empirical experiments, demonstrating the effectiveness of their proposed approach.

**Strengths:**

1. Overall paper is well-written and easy to read.
2. The proposed method uses stochastic variables to represent pseudo labels and estimates their
distributions. In addition, it learns variational beighbor labels to enhance the robustness of pseudo labels.
3. The method introduces a meta-generalization method to solve the problem of domain shifts.
4. The experiments conducted in the article are highly intuitive and sufficiently comprehensive.

**Weaknesses:**

1. Behind the Equation 2, the authors don’t explain the symbol delta.
2. Shoule include more ablations, such as pseudo-labeling with meta-generalization and meta-generalization with probabilistic pseudo-labeling.

**Questions:**

See above.

---

> ### Author Response · Authors · 2023-11-22
> **Response to reviewer Hsmr**
>
> We thank Reviewer Hsmr for their feedback.
>
> **delta symbol**
>
> The delta symbol indicates the Dirac delta distribution, also known as unit impulse. The approximation of $p(\theta_t|\theta_s, x_t)$ by $\delta(\theta_t=\theta_t^*)$ indicates we use the deterministic MAP value $\theta_t^*$ to approximate the distribution $p(\theta_t)$. We have clarified this in the updated paper.
>
> **More ablations for meta-generalization**
>
> Following your suggestion, we report additional ablations on PACS: meta-generalization with pseudo-labeling, and meta-generalization with probabilistic pseudo-labeling. Meta-generalization with the proposed variational pseudo-labeling improves the most. We have added these results in the experiments section.
>
>
> |                                                       | PACS  |
> |-------------------------------------------------------|-------|
> | Meta-generalization with pseudo-labeling              | 82.0  |
> | Meta-generalization with probabilistic pseudo-labeling | 82.7  |
> | Meta-generalization with variational pseudo-labeling  | 83.5  |

---

### Meta-Review · Area_Chair_YqJV · 2023-11-26

**Metareview:**

This paper proposes a test-time domain generation method from a probabilistic perspective by modeling pseudo labels as distributions. Specifically, variational neighbor labels are incorporated to generate more robust pseudo labels, and meta-learning-based algorithms are proposed to boost the performance. The experiments on seven datasets show its effectiveness.

**Strengths**

- The experiments conducted are highly intuitive and sufficiently comprehensive.

- The experiments conducted on seven datasets show its superiority over SOTA.

- The code is provided for reproducibility.

**Weaknesses**

- The experiments are limited to small-scale datasets. Some commonly adopted large-scale datasets are missing, e.g. ImageNet-C, VisDA.

- Many parts are not clear or seemingly not correct.

In sum, despite the good performance and intuitive experiments, the many parts in the paper are not clear.  In addition, the ablation study is not enough to prove the general effectiveness of the proposed modules. All the reviewers agree that this paper should be largely improved before publication. The AC thus recommends rejection to this paper.

**Justification For Why Not Higher Score:**

Many key parts are not well clarified and more results on larger datasets should be conducted. This paper should be undergone a major revision.

**Justification For Why Not Lower Score:**

N/A

---

### Decision · Program_Chairs · 2024-01-16

Reject